# Early Period of Training Impacts Adaptation for Out-of-Distribution Generalization: An Empirical Study

## Abstract

Prior research shows that differences in the early period of neural network training significantly impact the performance of in-distribution (ID) data of tasks. Yet, the implications of early learning dynamics on out-of-distribution (OOD) generalization remain poorly understood, primarily due to the complexities and limitations of existing analytical techniques. In this work, we investigate the relationship between learning dynamics, OOD generalization under covariate shift and the early period of neural network training. We utilize the trace of Fisher Information and sharpness, focusing on gradual unfreezing (i.e., progressively unfreezing parameters during training) as our methodology for investigation. Through a series of empirical experiments, we show that 1) changing the number of trainable parameters during the early period of training via gradual unfreezing can significantly improve OOD results; 2) the trace of Fisher Information and sharpness can be used as indicators for the removal of gradual unfreezing during the early period of training for better OOD generalization. Our experiments on both image and text data show that the early period of training is a general phenomenon that can provide Pareto improvements in ID and OOD performance with minimal complexity. Our work represents a first step towards understanding how early learning dynamics affect neural network OOD generalization under covariate shift and suggests a new avenue to improve and study this problem.

## 1 Introduction

Deep neural networks have achieved remarkable results on in-distribution (ID) data of a task they trained on but often performed poorly on out-of-distribution (OOD) data under input distribution shifts. OOD performance is critical for real-world applications, such as training on clean images or text but inferencing on noise-corrupted data (Hendrycks & Dietterich, 2019; Michel & Neubig, 2018), data obtained from different time periods (Lazaridou et al., 2021; Yao et al., 2022), across languages or domains (Wang et al., 2021; Talman & Chatzikyriakidis, 2019; Liu et al., 2022; Koh et al., 2021; Gulrajani & Lopez-Paz, 2021). Inadequate generalization to OOD settings is a key issue limiting the robustness and reliability of these models.

Prior research observed that variations in the early period of training have a significant impact on the model's ID performance (Golatkar et al., 2019; Achille et al., 2019; Mosbach et al., 2021; Fort et al., 2020) across scenarios including unimodal and multimodal settings when training from scratch, performing parameter-efficient fine-tuning, or using federated learning. The observation of such a period in diverse applications suggests that the early period of learning is generally important for neural network training (Kleinman et al., 2024), drawing parallels to biological phenomena like the critical learning period in animals (Achille et al., 2019; Kleinman et al., 2023).

In particular, intervening during the early period of training can significantly impact ID generalization at the end of training. Training techniques, such as adjusting optimization hyperparameters (e.g., weight decay, learning rate, or dropout; Golatkar et al. 2019; Jastrzebski et al. 2021; Mosbach et al. 2021; Liu et al. 2023b), using data augmentation (Golatkar et al., 2019; Liu et al., 2023c), or adding noise to weights (Frankle et al., 2020), impact learning dynamics early on and can significantly improve or degrade ID results depending on when they are applied or removed. Despite extensive

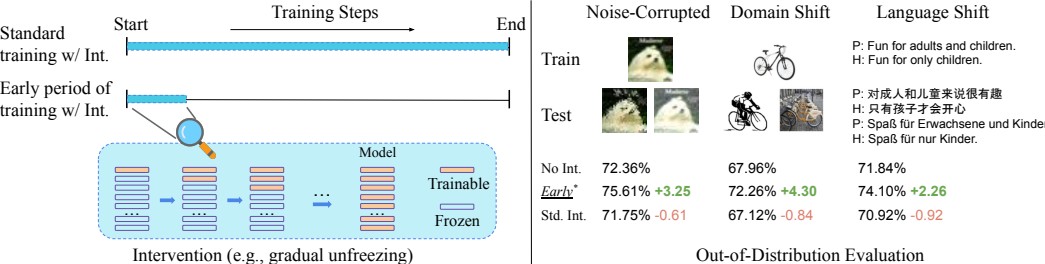

Figure 1: (Left) Interventions during the early period of training are applied for a much shorter time. (Right) Impact of intervention in the early period of training on OOD performance across diverse settings (CIFAR10, Krizhevsky 2009; Hendrycks & Dietterich 2019; Office-Home, Venkateswara et al. 2017; XNLI, Conneau et al. 2018). * indicates optimal OOD results (§5.1).

studies on early learning dynamics and ID generalization, to the best of our knowledge, the impact of the early training period on *OOD* generalization remains unexplored.

In this work, we focus on the impact of the early period of training on OOD generalization, specifically under the common input distribution shift (i.e., covariate shift, encompassing clean to noisy inputs, language, and domain shifts, etc.). We conduct a series of empirical investigations to explore this effect from a previously unexplored perspective – trainable parameters – by using gradual unfreezing (Howard & Ruder, 2018) to intervene in the early period of training (Figure 1). This method is a simple instance of broader training approaches with selective trainable parameters (Kumar et al., 2022; Lee et al., 2023), proven effective in adaptation for OOD generalization. We investigate changes in commonly used metrics to study generalization, namely Fisher Information and loss sharpness (Jastrzebski et al., 2021; Foret et al., 2021; Kwon et al., 2021; Zheng et al., 2021) during the early period of training, exploring their roles in shaping OOD generalization. Through targeted case studies, we demonstrate how leveraging the dynamics of the early period in training can be an effective strategy for "seizing" the moment for generalization across various scenarios.

To the best of our knowledge, we show for the first time that intervening through trainable parameters (i.e., gradual unfreezing) in the early period of training, can significantly enhance OOD generalization under covariate shift in various settings. Our results indicate that sharpness and Fisher Information metrics, though they may not be directly predictive of OOD generalization, can be used as indicators to optimize the timing of intervention removal for better OOD results. We validate this finding in both vision and language tasks, showing its ability to achieve Pareto improvements with minimal complexity. Our analysis and empirical evidence reveal new insights into how early learning dynamics impact neural network generalization, particularly under covariate shift, and suggest new avenues for studying OOD generalization.

## 2 RELATED WORK

**Early period of neural network training.** Under the standard usage of the term generalization (in-distribution, where training and testing data are assumed to be from the same distribution), prior work (Golatkar et al., 2019; Achille et al., 2019) shows that the early period of training of neural networks exhibits a "critical learning period" when trained from scratch. Regularization and interventions applied in this critical period affect final task results.

Jastrzebski et al. (2021) indicates that when learning with a lower learning rate, Fisher Information exhibits an "explosion" in the early period of training which impedes ID generalization. Applying regularization to the trace of Fisher Information alleviates the negative impact of the high Fisher Information. Liu et al. (2023c) shows the termination of MixUp (Zhang et al., 2018) early in training and switching to standard empirical risk minimization helps with better ID generalization. You et al. (2020); Frankle et al. (2020) shows that even winning "lottery tickets" emerge in the early period of training with large learning rates. The critical learning period is also found in many other settings, such as in multimodal models (Kleinman et al., 2023), in linear models (Kleinman et al., 2024), in transformers (Mosbach et al., 2021) and federated learning (Yan et al., 2022). However, these works only focus on ID generalization, neglecting the challenges of OOD generalization.

Prior work (Yang et al., 2024; Qiu et al., 2024) investigates into how models adapt to spurious correlations (which is a distinct form of OOD problem compared to the covariate shift examined in our work). These studies focus on the early formation of distinct spurious features during training and the mitigation strategies. However, the impact of trainable parameters (an increasingly important area in light of recent advancements in parameter efficiency and dynamic architectures) remains underexplored.

Kumar et al. (2022); Lee et al. (2023); Liu et al. (2023a) find that training different parts of a model at different times can alter learning dynamics and achieve better OOD results. Encouraged by these findings, we use gradual unfreezing (Howard & Ruder, 2018, a very simple form of training parts of a model at different times) as the main investigative tool in this paper. We focus on two key advancements: 1) more general settings (e.g., training from scratch, fine-tuning), and 2) the characterization of the early period of training and its relationship to OOD generalization.

**Fisher Information, sharpness and generalization.** Fisher Information has been studied in many prior works such as Chaudhari et al. (2017); Martens & Grosse (2015) to investigate and improve optimization behaviour. Similarly, sharpness is another popular metric used to study optimization behaviour and its relationship to generalization.

Jastrzebski et al. (2017) found a correlation between sharpness and the ratio of learning rate to batch size, which impacts generalization. Jiang et al. (2020); Dziugaite & Roy (2017); Neyshabur et al. (2017) provide theoretical backing for generalization error using sharpness-related measures and empirically show a correlation with generalization. While prior work believes that flatter (less sharp) minima in the loss landscape lead to better generalization in neural networks (Hochreiter & Schmidhuber, 1997; Keskar et al., 2017; Izmailov et al., 2018; Cha et al., 2021), there have been debates on whether sharp minima (such as a high largest eigenvalue of the training Hessian, $\lambda_{max}$) imply poor generalization (Dinh et al., 2017) and demonstrate the limits of $\lambda_{max}$ in explaining ID generalization (Kaur et al., 2023). Andriushchenko et al. (2023) demonstrate that adaptive sharpness is an unreliable metric for OOD generalization in the final solution.

Current research primarily examines the loss landscape at convergence to understand ID generalization. However, the role of Fisher information and sharpness metrics during early training and their relationship to final OOD generalization remains unclear.

## 3 PRELIMINARIES

We utilize Fisher Information (Fisher, 1925) and sharpness to analyze the training process. Below, we outline the specific metrics used in our experiments.

### 3.1 FISHER INFORMATION MATRIX (FIM)

Let $x$ be the inputs and $y$ be the labels of a dataset $D$. Given a neural network parameterized by $w$ with an output distribution $p_w(\cdot|x)$ for input $x$, the Fisher Information is defined as:

$$F(w) = \frac{1}{|D|} \sum_{x \in D} \mathbb{E}_{\hat{y} \sim P_w(\cdot|x)} \left[ \nabla_w \log p_w(\hat{y}|x) \nabla_w \log p_w(\hat{y}|x)^T \right]. \tag{1}$$

Note that $\hat{y}$ are sampled from $p_w(\cdot|x)$ and not equal to $y$ in general.

Fisher Information reflects the local curvature and measures the amount of information with respect to network parameters, i.e., how sensitive the network predictions are to the small changes in its parameters (Amari & Nagaoka, 2000). A higher value of an element of $F(w)$ indicates that a small change in the corresponding network parameter results in a significant change in the output, which can be interpreted as a "sharper" loss landscape.

Estimating the full $F(w)$ is generally expensive. Prior work shows that the trace of the Fisher Information, $\mathtt{tr}(\mathrm{F})$, correlates well with the full Fisher Information when used in real applications to capture signals during the learning process (Achille et al., 2019; Jastrzebski et al., 2021; Sung et al., 2021, inter alia). $\mathtt{tr}(\mathrm{F})$ is defined as

$$\mathtt{tr}(\mathrm{F}) = \frac{1}{|D|} \sum_{x \in D} \mathbb{E}_{\hat{y} \sim p_w(\cdot|x)} ||\nabla_w \log p_w(\hat{y}|x)||^2. \tag{2}$$

## 3.2 SHARPNESS

Let $\mathcal{L}_{\mathcal{D}}(w) = \frac{1}{|D|} \sum_{(x,y) \in D} \log p_w(y|x)$ be the loss over training datasets $D$, of a neural network parameterized by $w$, and $\delta$ be a small perturbation drawn from a noise distribution, such as a Gaussian distribution $\mathcal{N}(0, \rho^2 diag(c^2))$. The definitions of average and worst-case sharpness are (Foret et al., 2021; Kwon et al., 2021; Andriushchenko et al., 2023; Hochreiter & Schmidhuber, 1997):

$$S^{\rho}_{avg} = \mathbb{E}_{\delta \sim \mathcal{N}(0, \rho^2 diag(c^2))} \mathcal{L}_{\mathcal{D}}(w - \delta) - \mathcal{L}_{\mathcal{D}}(w), \tag{3}$$

$$S^{\rho}_{worst} = \max_{\|\delta \odot c^{-1}\|_p \leq \rho} \mathcal{L}_{\mathcal{D}}(w - \delta) - \mathcal{L}_{\mathcal{D}}(w), \tag{4}$$

where $\rho$ is a radius parameter of the noise, $c$ is a vector in the parameter space along which sharpness is measured and $\odot c^{-1}$ is element-wise multiplication.

Sharpness metrics measure how the loss changes with respect to small changes to model parameters.[1] While both the Fisher Information and sharpness are used for investigating loss landscapes and generalization, they offer different views (parameter space vs. loss) of the training process.

## 3.3 GRADUAL UNFREEZING

Gradual unfreezing (Howard & Ruder, 2018) progressively increases the number of trainable parameters (i.e., unfreeze, layer-by-layer) of a neural network from the top to the bottom of a network at a fixed interval of training steps, $k$ (i.e., the unfreezing interval). In this paper, we use a modified formulation of gradual unfreezing (Liu et al., 2023a), where we progressively unfreeze "blocks" of parameters during the early period of training top-down (a block of parameters can range from a single layer to several consecutive layers). In our experiments, we use the namespace of the parameters used in standard implementations to determine blocks. See Appendix B for the algorithm. This method, along with the top-down unfreezing, is chosen as the analysis tool due to its proven effectiveness in achieving state-of-the-art performance across various transfer learning settings (Howard & Ruder, 2018; Kumar et al., 2022; Liu et al., 2023a; Reinhardt et al., 2024).

## 4 EXPERIMENTAL SETUP

We study three experimental settings in this work, covering a diverse set of tasks and scenarios including training from scratch then inference with noise-corrupted images, fine-tuning a pre-trained model for domain generalization, and parameter-efficient fine-tuning for language shift generalization. All experimental results are averaged over 6 runs (for MNIST due to high variances) or 4 runs (all other datasets) and only ID data is used for model selection. See Appendix C and Appendix D for details on the evaluation datasets and hyperparameters.

**Training from scratch, noise-corrupted input shift.** In this setting, we train a ResNet18 (He et al., 2016) from scratch using the MNIST (LeCun et al., 1998), CIFAR10 (Krizhevsky, 2009), or CIFAR100 (Krizhevsky, 2009). For OOD evaluation, we use the corrupted corresponding evaluation datasets, MNIST-C (Mu & Gilmer, 2019), CIFAR10-C (Hendrycks & Dietterich, 2019) and CIFAR-100-C (Hendrycks & Dietterich 2019, averaging results across corruption types and severities. ID evaluation is done on the original test sets.

**Fine-tuning from a pre-trained model, domain shift.** Here, we fine-tune an ImageNet (Deng et al., 2009) pre-trained vision transformer (ViT, Wu et al. 2020). We use two popular domain shift datasets, namely Office-Home (Venkateswara et al., 2017) and DomainNet (Peng et al., 2019) for evaluation. We use a single source-domain for training, evaluating all other domains that are not part of the training for Office-Home. For DomainNet, we train on the three domains with the least data (for efficiency reasons) and evaluate on the test sets of all other domains that are not the same as the training domain (i.e., $\text{Domain}_{\text{train}} \in \{\text{Sketch, Infograph, Clipart}\}$, $\text{Domain}_{\text{test}} \in \{\text{Sketch, Infograph, Clipart, Real, Painting, Quickdraw}\}$, see Appendix C).

---

[1]The sharpness can be negative.

**Parameter-Efficient Fine-Tuning (PEFT) with a pre-trained model, language shift.** We also conduct experiments using a language transformer. Since pre-training and fine-tuning are common for adapting foundational models, we examine the cross-lingual transfer (train with English data, test with other languages) task using PEFT with the LoRA (Hu et al., 2022) adapters. Here, ID data refers to English task data (for training and validation), while OOD data are in other languages. We train with SQuAD (Rajpurkar et al. 2016, English, question and answering task) and MNLI (Williams et al. 2018, English, natural language inference task), and evaluate on XQuAD (Artetxe et al., 2020), MLQA (Lewis et al., 2020) and XNLI (Conneau et al., 2018). We use XLM-RoBERTa (Conneau et al., 2020) as the pre-trained multilingual transformer backbone.

**Learning dynamics metrics.** We use $\rho = 0.01$ to calculate the sharpness (both average-case and worst-case) with 15 noise samples (it is computationally expensive to use a larger number of noise samples), and $L2$ norm for the worst-case sharpness. We normalize the $\text{tr}(\text{F})$ by the number of trainable parameters. We use the Auto-PGD algorithm (Croce & Hein, 2020) as implemented in Andriushchenko et al. (2023) (we refer the readers to the original papers for details) for computing worst-case sharpness, as it is a hyperparameter-free estimation method.

# 5 IMPACT AND TIME-SENTISITIVITY OF EARLY INTERVENTIONS ON OUT-OF-DISTRIBUTION GENERALIZATION

## 5.1 TIMING IS CRITICAL FOR REMOVING INTERVENTIONS

Here, we investigate how unfreezing interval $k$ affects ID and OOD results. For all experiments in this section, the smallest $k$ is 1 (unfreeze a block of parameters every one batch update) and the largest $k$ is determined by equally dividing the total training steps among all blocks (Howard & Ruder, 2018; Raffel et al., 2022; Kumar et al., 2022; Lee et al., 2023). We show the relative change in the test results (by subtracting the test results using gradual unfreezing to the results using standard training).

**Training from scratch, noise-corrupted input shift.** Figure 2 shows the relative change in test results compared to standard training. Gradual unfreezing of trainable parameters significantly impacts OOD generalization (i.e., noise-corrupted images) as early as after a single batch of data, and this effect is particularly pronounced in simpler datasets like MNIST. Extending the unfreezing interval during training initially has minimal impact on ID performance, but later leads to a significant decline, especially at a faster rate with CIFAR. The observed deterioration in ID performance over extended unfreezing intervals mirrors trends from early-stage training interventions and aligns with previous findings (Golatkar et al., 2019; Achille et al., 2019) using other interventions.

The influence of gradual unfreezing on OOD results serves as evidence for the importance of early training periods for OOD generalization. Notably, gradual unfreezing reveals a trade-off between ID and OOD performance for CIFAR datasets, with a brief window where OOD results improve before a sharp decline in ID performance. These results suggest that there may be a critical range of $k$ during early training where ID performance remains stable and OOD performance improves. This time-sensitive observation persists over different learning rates (Figure 2(b)) and model depths (Figure 7 in Appendix F).

**Fine-tuning from a pre-trained model, domain shift.** Figure 3(a) presents the relative change in domain generalization performance when fine-tuning a pre-trained ViT on single source-domain data. Consistent with previous observations on noise-corrupted input images, results on DomainNet and Office-Home both exhibit a time-sensitive nature in parameter training. [2] Notably, there is also a specific period during training where domain generalization results improve significantly (+2.72% points in accuracy for DomainNet and +4.30% points for Office-Home) with minimal impact on ID evaluation results (DomainNet).

**PEFT with a pre-trained model, language shift.** We continue to observe a consistent pattern in OOD results (Figure 3(b)) with the prior two scenarios. However, the ID performance for SQuAD also shows improvements. In particular, unfreezing around 1000 and 1600 steps obtain high improvements on average test F1 scores (+2.25% on XNLI and +1.73% on SQUAD). Similar to prior observations, both ID and OOD results are poor when unfreezing occurs later in the training process.

---

[2]Since there is no official test set for Office-Home, the ID evaluation results are omitted.

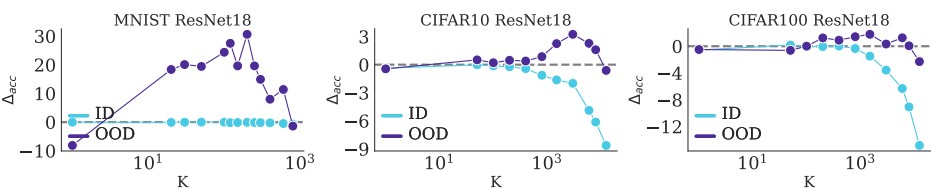

(a) Maximum OOD accuracy improvements are 30.63%, 3.25%, and 1.78% points in accuracy.

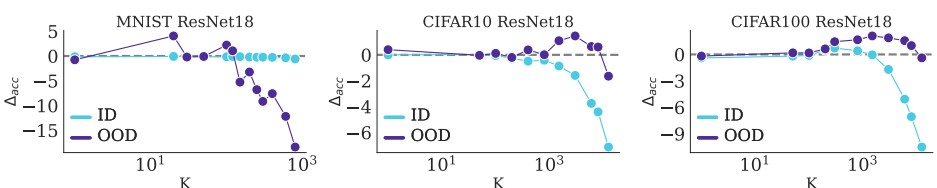

(b) Maximum OOD accuracy improvements are 4.06%, 1.46%, and 2.10% points in accuracy, using 1/10th of the learning rate as in sub-figure (a).

Figure 2: Changes in ID and OOD (noise-corrupted images) evaluation results when unfreezing parameters at different times (i.e., $k$) highlight the early training period's impact on OOD generalization. $\Delta_{acc}$ is calculated by subtracting gradual unfreezing results from standard training. The x-axis is in the log scale. Each data point on the plot is obtained by averaging over 6 runs for MNIST and 4 runs for CIFAR datasets (a total of 166 experiments per subfigure).

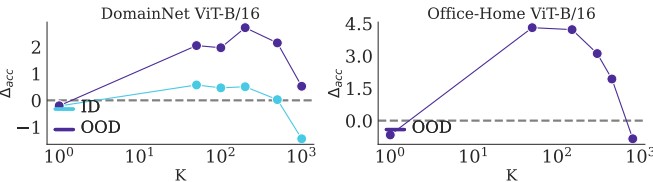

(a) The maximum improvements in OOD (domain shift) results are +2.72% in accuracy for Domain-Net and +4.30% points in accuracy for Office-Home, averaged over 4 runs.

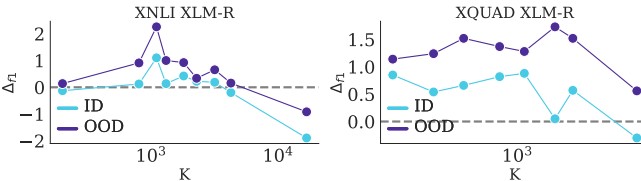

(b) The maximum improvements in OOD (language shift) results are +2.26% points on XNLI and +1.73% points on SQUAD in F1, averaged over 4 runs.

Figure 3: Changes in ID and OOD evaluation results when unfreezing parameters at different times (i.e., $k$) for domain shift (vision) and language shift (text) with pre-trained transformers. $\Delta$ is calculated by subtracting the gradual unfreezing results from standard training, averaged over 4 runs (168 experiments in total for subfigure (a) due to single source-domain training and 68 experiments for subfigure (b)). The x-axis is in the log scale.

**Summary of findings.** The early period of training impacts adaptation to OOD data under covariate shifts. Intervening during this period with gradual unfreezing leads to a time-sensitive trade-off between OOD and ID performance. While factors like data quality or sophisticated learning objectives could influence OOD performance, our results suggest that early, well-timed intervention on trainable parameters also significantly influences models' eventual performance, making this phase a key target for improving OOD results at minimal complexity.

## 5.2 DISCUSSIONS

**Why does gradual unfreezing help OOD?** Kumar et al. (2022, linear-probing then fine-tuning) proposes a method that aligns the classification head (while other parameters kept frozen) with ID data early in training to prevent feature distortion during fine-tuning, leading to better OOD generalization. We suspect that gradual unfreezing could be exploiting a similar mechanism during the early period of training when fine-tuning from a pre-trained model.

Let $\mathbf{g}_b$ denote the gradient of a mini-batch in the training set $b \in \mathcal{B}$. Let $\hat{\mathbf{g}}$ denote the full-batch gradient for the training set. Define the (average) *gradient similarity* at each training step by $\text{GS} \triangleq \frac{1}{|\mathcal{B}|} \sum_{b \in \mathcal{B}} \frac{\mathbf{g}_b \cdot \hat{\mathbf{g}}}{\|\mathbf{g}_b\|\|\hat{\mathbf{g}}\|}$ where ($\cdot$) denotes vector multiplication. Tracking GS during training, we find that when training from scratch, GS is higher when using gradual unfreezing than standard training during the early period of training. The difference in GS disappears after the early period (Figure 4 shows GS for the classification head, additional layers in Appendix H.1). This suggests that gradual unfreezing could help to better align early mini-batch gradients to the full-batch gradient.

Improving alignment could prevent overfitting to specific mini-batches and reduce learning spurious features, especially early in training, where such features can have lasting deleterious effects. To verify, we conduct additional experiments using the WaterBirds dataset (Sagawa et al., 2020, commonly used for spurious correlation study). We find that gradual unfreezing indeed improves worst-group accuracy over standard training (see Appendix F.4).

**Other interventions during the early period of training.** Beyond gradual unfreezing, we found that other interventions also exhibit the time-critical nature of training for OOD generalization. Currently, the list of interventions includes learning rate warm-up and delaying the application of a regularizer that minimizes $\mathtt{tr(F)}$ (Jastrzebski et al., 2021). We refer the reader to Appendix F.2 and F.3 for details and additional results. While the gain in OOD generalization for other interventions is less significant than restraining trainable parameters (i.e., through gradual unfreezing), these additional cases indicate that the time-critical nature of removing/applying intervention for OOD generalization is a general phenomenon. We will investigate them in detail in the future.

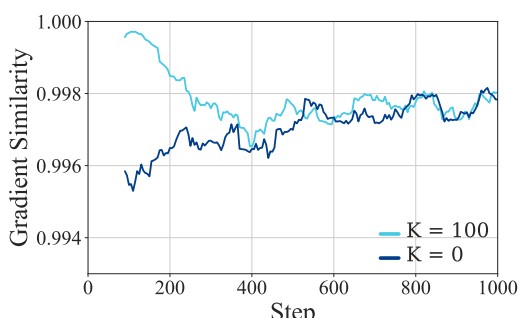

Figure 4: Gradient similarity (mini-batch vs full-data) for the classification head of a ResNet18 trained with CIFAR10. The mini-batch gradient is more similar to the full-data gradient in the early period of training when gradual unfreezing is applied (K=100) compared to standard training (K=0).

## 6 CAN WE USE LEARNING DYNAMICS TO "SEIZE" THE EARLY PERIOD OF TRAINING FOR OOD GENERALIZATION?

### 6.1 LEARNING DYNAMICS ANALYSIS

To analyze the characteristics of the early period of training with gradual unfreezing, we examine the learning dynamics using the three metrics described in §3.

Figure 5(a) and Figure 5(c) show the learning dynamics for the early period of training from scratch or with PEFT (in both cases, only randomly initialized parameters are updated). We see that by initially freezing and subsequently gradually unfreezing the trainable parameters, we induce higher Fisher Information and $S_{avg}^{\rho}$, $S_{worst}^{\rho}$ at the beginning of training compared to standard training. In general, the longer we withhold parameters, the higher the level of sharpness and $\mathtt{tr(F)}$ we can sustain. Unfreezing parameters reduce these metrics.

Figure 5(b) (domain shift, fine-tuning a pre-trained backbone) shows some inconsistency in the metrics initially. However, as parameters are withheld longer, sharpness and $\mathtt{tr(F)}$ sustain. Note

that, unlike the previous cases, this experiment directly fine-tunes a pre-trained model without adding randomly initialized parameters.

While $S_{avg}^{\rho}$, $S_{worst}^{\rho}$, and $\mathrm{tr(F)}$ differ in definition, they are all sensitive to the early period of training.[3] We identify a pattern consisting of two phases: 1) an initial phase of rapid change (e.g., before the first 50-100, 1000 or 2000 steps in the three subfigures in Figure 5 respectively), and 2) a subsequent stabilization phase where the rate of change of the metric decreases.

**Summary of findings.** *1)* Gradual unfreezing alters learning dynamics during the early period of training, as measured by the metrics in §3. *2)* The inconsistent learning dynamics across metrics when fine-tuning a pre-trained backbone suggest that sharpness alone does not reliably predict OOD generalization in the modern transfer learning setup. This points to the need to develop new theoretical metrics in different OOD scenarios. Empirically, low sharpness during *early training period* does not guarantee optimal OOD results (e.g., GU increases sharpness early on when training from scratch, yet yields better OOD results empirically), despite the recent success of many sharpness minimization methods for ID. *3)* In standard training (without interventions), metrics exhibit two phases: an initial phase of rapid change, followed by a stabilization phase.

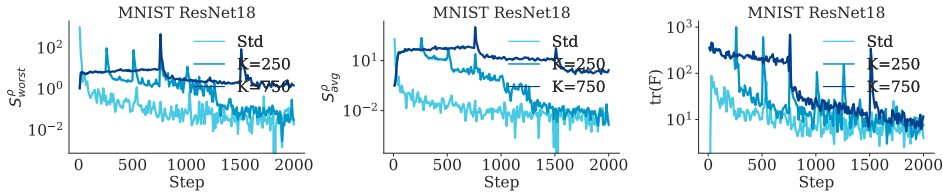

(a) Training from scratch: the plot shows metrics when unfrozen at steps $k = \{250, 750\}$ compared to standard training. The best OOD result in this plot is when $k = 250$ (+19.62% points compared to standard training). We also observe similar trends with 1/10 of the learning rate here (see Appendix H.4).

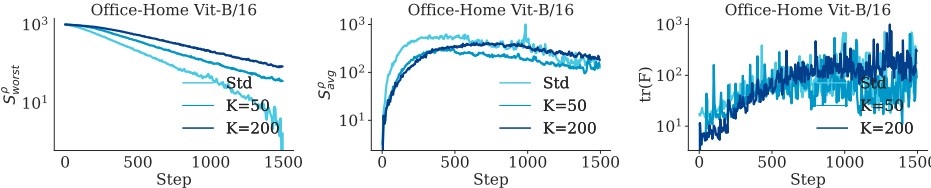

(b) Fine-tuning from a pre-trained ViT model on Office-Home (Art as the source domain for training): the plot shows metrics when unfrozen at steps $k = \{50, 200\}$ compared to standard training. The best OOD result in this plot is when $k = 50$ (+4.30% points compared to standard training).

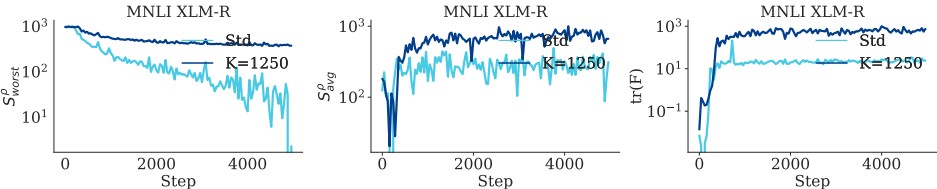

(c) Fine-tuning of a text Transformers with LoRA adapters: the plot shows metrics when unfrozen at steps $k = \{1250\}$ compared to standard training.

Figure 5: Learning dynamics with three metrics: $\mathrm{tr(F)}$, $S^{\rho}avg$, and $S^{\rho}worst$. Unfreezing parameters at different steps impact early learning dynamics. The y-axis is log-scaled and normalized between 0 and 1000 for clarity.

---

[3]See Appendix H for additional learning dynamics with similar trends.

## 6.2 Do learning dynamics signal the right time for intervention removal?

**Learning dynamics criteria for improved OOD generalization.** Here, we investigate the learning dynamics of ResNet18 on MNIST when training from scratch. We combined our findings in §5.1 and §6.1 to arrive at the hypothesis that the optimal range of $k$ for achieving the best overall ID and OOD performance is after the initial rapid change of sharpness and $\mathtt{tr}(\mathrm{F})$ but not too long afterward. For instance, the ID results deteriorated rapidly after 800-1000 training steps for the CIFAR datasets in Figure 2, while OOD results were still improving.

This observation suggests that the optimal time to remove intervention (i.e., unfreeze parameters in our case) while maintaining ID results (less than 0.5 points decrease in accuracy) and achieving better OOD results should meet two specific criteria: **1)** after the initial rapid change of sharpness or $\mathtt{tr}(\mathrm{F})$, and **2)** before the stabilization phase progresses too far.

Criterion **2)** is evident across all figures in our prior experiments in §5, as larger values of $k$ consistently degrade both ID and OOD performance. To assess the criterion **1)**, we focus on the MNIST dataset and identify $\hat{k}$ as the earliest ending step of the initial rapid-changing phase among the three metrics ($S^\rho_{worst}$, $S^\rho_{avg}$ and $\mathtt{tr}(\mathrm{F})$). We then experiment with 10 different $k$ values, spaced 10 steps apart, both less and greater than $\hat{k}$. For $k < \hat{k}$, we obtained median OOD (ID) accuracies of 52.72 (98.93). For $\hat{k} < k$, we obtained median OOD (ID) accuracies of 53.54 (98.91). This result helps to validate the first criterion since the median OOD accuracy is lower for $k < \hat{k}$ with minimal change in ID accuracy. Together, this analysis suggests that the stabilization of metrics after the initial phase could be a useful signal to determine the optimal **time** to introduce new trainable parameters.

**Hypothesis validation with a heuristic algorithm.** To further validates our hypothesis, we use a heuristic algorithm that satisfies the above-mentioned criteria to determine the stabilization time of the three metrics (we first detect a significant change in metrics, then detect the stabilization point of the metrics, the algorithm is given in Appendix E). The OOD results are then compared with ten random sampled $k$ values per dataset to determine the winning rate (i.e., the percentage of times when the value picked by the algorithm is better than a randomly sampled value).

Training from scratch, noise-corrupted input shift. As shown in Table 1, using a heuristic algorithm is better than performing a random hyperparameter search the majority of the time. In most cases, the degradation of ID accuracy is within 0.5 percentage points. This further validates that the stabilization of $S^\rho_{worst}$, $S^\rho_{avg}$ and $\mathtt{tr}(\mathrm{F})$ could signify the removal of interventions (in our case gradual unfreezing) to trade-off a small amount of ID performance for better OOD results. While $\mathtt{tr}(\mathrm{F})$ shows better results, there isn't a clear winning metric for intervention removal due to: 1) the metrics exhibit high noise during training, and 2) the stabilization points determined by different metrics either match or are very close to each other. We defer the exploration of more sophisticated algorithms to future work. Nevertheless, our experiments show that an optimal intervention window exists that can effectively balance good ID and ODD results and the stabilization of sharpness and $\mathtt{tr}(\mathrm{F})$ could signal the right time to remove interventions.

Table 1: Results using the heuristic algorithm to find $\hat{k}$ for gradual unfreezing (GU). Best OOD results are bolded. The algorithm can determine the same value of $\hat{k}$ in different metrics in multiple cases (hence the same results). WR stands for winning rate (OOD). See Appendix E for visualization of $\hat{k}$ overlay on the learning dynamics.

| Method | MNIST RN18 ID / OOD | CIFAR10 RN18 ID / OOD | CIFAR100 RN18 ID / OOD | WR - |
|---|---|---|---|---|
| Standard | $99.06_{\pm0.08}/33.36_{\pm10.81}$ | $93.32_{\pm0.23}/72.36_{\pm0.63}$ | $71.07_{\pm0.36}/45.10_{\pm0.39}$ | - |
| $\mathrm{GU}_{S^\rho_{worst}}$ | $98.78_{\pm0.15}/52.48_{\pm7.70}$ | $93.06_{\pm0.06}/72.75_{\pm0.84}$ | $70.68_{\pm0.18}/45.19_{\pm0.62}$ | 60% |
| $\mathrm{GU}_{S^\rho_{avg}}$ | $98.78_{\pm0.15}/52.48_{\pm7.70}$ | $93.02_{\pm0.05}/72.58_{\pm0.49}$ | $70.67_{\pm0.20}/45.35_{\pm0.60}$ | 60% |
| $\mathrm{GU}_{\mathtt{tr}(\mathrm{F})}$ | $98.91_{\pm0.26}/\mathbf{54.12}_{\pm10.23}$ | $93.02_{\pm0.10}/\mathbf{73.56}_{\pm0.45}$ | $70.78_{\pm0.31}/\mathbf{45.82}_{\pm0.56}$ | 83% |

PEFT with a pre-trained model, language shift. Results using $\mathtt{tr(F)}$ to determine $\hat{k}$ for gradual unfreezing are shown in Table 2 ($\hat{k}$ values are in Appendix E, results with $S_{worst}^{\rho}$ / $S_{avg}^{\rho}$ are in Table 6 in the Appendix) and the winning rate is 80%. The ID results are not sacrificed in this experimental setting, hence further pointing towards that the stabilization of sharpness and $\mathtt{tr(F)}$ could signify 'when' to remove intervention in the early period of training for better OOD generalization.

Table 2: Cross-lingual transfer results of standard training and using $\mathtt{tr(F)}$ to determine unfreezing interval $\hat{k}$ for gradual unfreezing (GU), best OOD results are bolded. WR stands for winning rate, averaged over 10 randomly sampled $k$ per training dataset. EM is the exact match score.

| Method | XQuAD F1- En/X-ling | EM- En/X-ling | MLQA F1- X-ling | EM- X-ling | XNLI Acc- En/X-ling | WR |
|---|---|---|---|---|---|---|
| Standard | $82.96_{\pm0.49}/68.72_{\pm0.85}$ | $71.39_{\pm0.25}/52.64_{\pm0.66}$ | $56.27_{\pm0.80}$ | $40.93_{\pm0.55}$ | $83.17_{\pm0.29}/71.84_{\pm0.52}$ | - |
| $\mathrm{GU}_{\mathtt{tr(F)}}$ | $83.77_{\pm0.57}/\mathbf{70.70}_{\pm0.27}$ | $72.33_{\pm0.69}/\mathbf{54.40}_{\pm0.27}$ | $\mathbf{58.47}_{\pm0.21}$ | $\mathbf{42.31}_{\pm0.17}$ | $83.36_{\pm0.13}/\mathbf{72.49}_{\pm0.42}$ | 80% |

Fine-tuning from a pre-trained model, domain shift. The domain generalization results on Domain-Net, using $\mathtt{tr(F)}$ to determine $\hat{k}$ for gradual unfreezing, achieve an accuracy of 37.86%, compared to 35.34% with standard training, with a winning rate of 90%. The complete results are in Table 7 in appendix. Once again, our results align with the trends observed in the previous two cases.

**Summary of findings.** Our case studies show that the learning dynamics can be effective indicators for determining the optimal timing for intervention removal, with $\mathtt{tr(F)}$ being a slightly better metric overall (when trainable parameters are randomly initialized). While the improvement in OOD performance may be modest, this higlights the connection between learning dynamics and OOD generalization, as well as its potential applications. Since our understanding of the relationship between learning dynamics and OOD generalization is nascent, we hope this initial work will encourage further exploration in this area.

## 7 CONCLUSIONS

In this work, we investigate the early period of training and its impact on OOD generalization under covariate shift. We show that interventions by altering trainable parameters (i.e., progressively changing the number of trainable parameters through gradual unfreezing) during the early period of training improve OOD generalization. This is validated across various vision and language tasks, achieving Pareto improvements with minimal complexity. We emphasize the overlooked role of trainable parameters during the early period of training. Unlike prior work on ID generalization, we empirically observed that sharpness and $\mathtt{tr(F)}$ during the early period of training may not be indicative of the OOD generalization, but can be indicative of "when" to remove interventions.

In light of these findings, it is also essential to consider the broader context of the training strategy studied in this work. The significance of methods that modify only parts of the final model, along with the growing focus on efficient training and fine-tuning — such as freezing parameters (e.g., Adapters, Houlsby et al. 2019; Pfeiffer et al. 2020; Hu et al. 2022) or dynamic architectures (Yoon et al., 2018; Evci et al., 2022; Gu et al., 2021) — cannot be overstated. Our findings contribute to a deeper understanding of the early period of training and OOD generalization, and suggest new research directions, including the development of theoretical metrics to better predict OOD generalization.

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

## A LIMITATIONS

There are several limitations to our work. We empirically evaluate three model architectures, and three types of covariate shift, and our selection may not encompass all possibilities and real-world applications. However, we believe our insights have generalizable value. Our empirical observations reveal correlations between changes in training dynamics and OOD generalization. Future research could explore causal interventions to better understand and enhance this relationship.

In our work, we utilize a hand-crafted algorithm to determine metric stabilization time, aiming to show that sharpness stabilization could indicate the time needed for the removal of interventions. However, leveraging training dynamics during the early period of training requires additional computation compared to standard training. While this is not a concern for our study, as our focus is on uncovering insights, more efficient, theory-driven algorithms and metrics should be explored for future practical applications.

## B GRADUAL UNFREEZING

Following the notations and algorithm in Liu et al. (2023a), let FORWARD($*$) be the standard forward pass, and BACKWARD($*$) calculates gradients and performs updates for trainable parameters. The modified gradual unfreezing algorithm is in Algorithm 1.

In our experiments, we partition the blocks by their natural namespaces as follows:

**ResNet18:** The definition block follows the standard implementation of ResNet, with an input convolution layer and a batch norm group together as the additional block. The model parameters are partitioned into 5 blocks, and a classification head.

**VGG11:** The definition block follows the standard implementation of VGG, with 8 blocks in total. The classification head consists of 3 linear layers with a ReLU function in between. The results are in the Appendix.

---

**Algorithm 1** Gradual Unfreezing

---

**Require:** A model's eventual trainable parameters are partitioned into blocks $j \in \{0, \ldots, L-1\}$ parameterized by $\theta_j$, with a task-specific classification head $C$, and an unfreezing interval $k$. A set $\mathcal{S}$ of the indices of parameter blocks to unfreeze.

1: Initialize $C, \theta_j$ for all $j$
2: $\mathcal{S} \leftarrow \{C\}$
3: $j \leftarrow L - 1$
4: **for** $i = 1 \ldots N$ **do**
5:     Sample a data batch $b \sim D$
6:     **if** $i \mod k == 0$ **and** $i \leq kL$ **then**
7:         $\mathcal{S} \leftarrow \mathcal{S} \cup \{\theta_j\}$
8:         $j \leftarrow j - 1$
9:     **end if**
10:    FORWARD($*$)
11:    BACKWARD($\mathcal{S}$)
12: **end for**

---

**XLM-RoBERTa + LoRA:** The experiment follows Liu et al. (2023a). Each parameter block consists of 2 sets of LoRA adapters added to the query and value of the backbone transformer from the same layer. The LoRA parameters are partitioned into 12 blocks, and a classification head, where the classification head and the last layer of LoRA adapters are trainable initially.

## C DATASETS

We provide additional information on the datasets used for evaluation in our experiments.

**MNIST-C** (Mu & Gilmer, 2019), **CIFAR10-C/CIFAR100-C** (Hendrycks & Dietterich, 2019): This is the noise-corrupted version of the classic image classification datasets MNIST/CIFAR10/CIFAR-100. There are 15 different corruptions in the evaluation dataset, namely frost, fog, gaussian blur, gaussian noise, glass blur, impulse noise, jpeg compression, motion blur, pixelate, saturate, shot noise, snow, spatter, speckle noise, and zoom blur, across 5 severity levels. There are a total of 10 classes each for MNIST-C/CIFAR10-C, and 100 classes for CIFAR100-C.

**Office-Home** (Venkateswara et al., 2017): This is an image classification task where the images are organized into four different domains: Clipart, Art, Photo and Real. There are a total of 65 classes for classification in this dataset. We considered four domain transfer settings: from Clipart(C) to Art(A)/Photo(P)/Real(R); from A to C/P/R; from P to A/C/R; and from R to A/P/C.

**DomainNet** (Peng et al., 2019): This is an image classification task where the images are organized into six different domains: Infograph, Sketch, Real, Quickdraw, Painting, and Clipart. There are a total of 345 classes for classification in this dataset. Due to resource constraints and efficiency, we considered three transfer settings (with the least amount of training data): from Infograph(I) to Sketch(S)/Real(R)/Quickdraw(Q)/Painting(P)/Clipart(C); from C to I/S/R/Q/P; and from S to I/R/Q/P/C.

**XQuAD** (Artetxe et al., 2020): This is a parallel dataset for evaluating cross-lingual question answering, with an evaluation set covering 11 languages (excluding English): Arabic, German, Greek, Spanish, Hindi, Russian, Thai, Turkish, Vietnamese, Chinese, Romanian. The task is the classify the start and end of the answer given a question and a context.

**MLQA** (Lewis et al., 2020): This is a highly parallel dataset for evaluating cross-lingual question answering. The dataset consists of an evaluation set covering 6 languages (excluding English): Arabic, German, Spanish, Hindi, Vietnamese and Simplified Chinese. The task is the classify the start and end of the answer given a question and a context.

**XNLI** (Conneau et al., 2018): This is a multilingual natural language inference dataset covering 14 languages (excluding English): French, Spanish, German, Greek, Bulgarian, Russian, Turkish, Arabic, Vietnamese, Thai, Chinese, Hindi, Swahili and Urdu. The task is to classify a pair of sentences as having either an entailment, contradiction or neutral relationship.

## D  HYPERPARAMETERS

The hyperparameters are listed in Table 3 for our experiments. We use the default hyperparameters for the AdamW optimizer, except for the learning rate. All other hyperparameters for the transformer experiments follow Liu et al. (2023a), and we use the HuggingFace PEFT (Mangrulkar et al., 2022) implementations of LoRA. We report results over 6 random seeds for MNIST (due to the high variance in OOD results), and we use 4 random seeds for all other experiments. Standard data augmentation techniques are applied across all experiments. For the CIFAR datasets, we use random cropping and horizontal flipping. For the domain shift datasets, we apply resizing and cropping, horizontal flipping, colour jittering, and grayscaling, following the approach in Gulrajani & Lopez-Paz (2021). The experiments use a single NVIDIA P100, A6000 or A100 GPU depending on the availability.

For our domain shift experiments, we used the total training steps and conducted evaluations across various settings with a single source-domain training (hyperparameter determined following Gulrajani & Lopez-Paz 2021).

For calculating $S_{worst}^{\rho}$ and $S_{avg}^{\rho}$, we use $L2$ norm and $\rho = 0.01$ with 15 examples. We follow the setup in Andriushchenko et al. (2023) and use the implementation with 2048 data points from the training data (un-augmented when calculating sharpness metrics) for all experiments. We use a batch size of 256, except for SQuAD (the batch size is 32) for calculating all the metrics. The sharpness and $\mathrm{tr}(\mathbf{F})$ are recorded every 10 batches (steps) for all datasets.

Table 3: Hyperparameters used in our experiments.

| | MNIST RN18 | CIFAR10 RN18 | CIFAR10 VGG11 | CIFAR100 RN18 | SQuAD XLM-R | MNLI XLM-R | Office-Home Vit-B/16 | DomainNet Vit-B/16 |
|---|---|---|---|---|---|---|---|---|
| optimizer | AdamW | SGD | SGD | SGD | AdamW | AdamW | AdamW | AdamW |
| lr scheduler | const. | const. | const. | const. | linear | linear | const. | const. |
| $lr_d$ | 0.01 | 0.1 | 0.15 | 0.01 | 0.0005 | 0.0005 | 0.00005 | 0.00005 |
| batch size | 128 | 128 | 128 | 128 | 32 | 128 | 128 | 128 |
| training epochs | 10 | 200 | 200 | 200 | 15 | 15 | - | - |
| training steps | - | - | - | - | - | - | 5000 | 15000 |
| weight decay | 0.01 | 0 | 0 | 0.0005 | 0.01 | 0.01 | 0.01 | 0.01 |
| momentum | 0.9 | 0 | 0 | 0.9 | - | - | - | - |
| LoRA r | - | - | - | - | 8 | 8 | - | - |
| LoRA alpha | - | - | - | - | 8 | 8 | - | - |
| LoRA dropout | - | - | - | - | 0.2 | 0.2 | - | - |

# E ALGORITHM TO DETERMINE THE UNFREEZE TIME

To verify our hypothesis, we use a simple algorithm with a heuristic to determine the unfreezing interval $\hat{k}$, Algorithm 2 presents the flow, $\tau$ is 3 or 8 and $\epsilon$ is 0.02 (i.e., the percentage of change in the signal is within 2%). The algorithm takes $t_{\Delta_{\hat{S}}}$ as the input, which is the index marking the end of the rapid increase of the signal using a similar logic.

---

**Algorithm 2** Find Stabilization

---

1: **procedure** FIND_STABILIZATION_BY_MEAN($\hat{S}, t_{\Delta_{\hat{S}}}, \tau, \epsilon$) ▷ $\hat{S}$ is an array of normalized signal when only the head is trainable, $t_{\Delta_{\hat{S}}}$ is the index marking the end of the rapid increasing of the signal, $\tau$ is the window for smoothing the signals, $\epsilon$ is the threshold in changes of the signal for stabilization.
2:  **if** $t_{\Delta_{\hat{S}}} > 0$ **then**
3:      $\hat{S} = \hat{S}[t_{\Delta_{\hat{S}}}:]$
4:  **end if**
5:  $\mu_{\hat{S}}$ = moving_average($\hat{S}, \tau$)
6:  $\Delta_{\mu_{\hat{S}}}$ = np.abs(np.diff($\mu_{\hat{S}}$))
7:  **for** i, $\delta$ in enumerate($\mu_{\hat{S}}$) **do**
8:      **if** $\delta \leq \epsilon$ **then**
9:          index = i                                    ▷ The first time where the change is smaller than $\tau$.
10:         break
11:     **end if**
12: **end for**
13: **if** $t_{\Delta_{\hat{S}}} > 0$ **then**
14:     index = index + $t_{\Delta_{\hat{S}}}$
15: **end if**
16: return index
17: **end procedure**

---

Using the heuristic algorithm, we determine the value $\hat{k}$ for experiments in Table 4, where we observe the determined $\hat{k}$ are very close to each other except for VGG with CIFAR10 and XLM-R with SQuAD. All the $\hat{k}$ values are shown visually in Figure 6, overlaying on top of the learning dynamics.

Table 4: Different $k$ determined by Algorithm 2.

| Metric | MNIST RN18 | CIFAR10 RN18 | CIFAR100 RN18 | CIFAR10 VGG11 | SQuAD XLM-R | MNLI XLM-R |
|---|---|---|---|---|---|---|
| $S^\rho_{worst}$ | 270 | 230 | 260 | 960 | 810 | 780 |
| $S^\rho_{avg}$ | 270 | 270 | 250 | 1010 | 1090 | 720 |
| $\mathtt{tr(F)}$ | 210 | 260 | 230 | 250 | 1310 | 790 |

Table 5: Results using the heuristic algorithm (Appendix E) to find $\hat{k}$ for gradual unfreezing (GU), best OOD results are bolded. The algorithm can determine the same value of $\hat{k}$ in different metrics in multiple cases (hence the same results). WR stands for winning rate (OOD).

| Method | MNIST RN18 ID / OOD | CIFAR10 RN18 ID / OOD | CIFAR100 RN18 ID / OOD | WR - | CIFAR10 VGG11 ID / OOD | WR |
|---|---|---|---|---|---|---|
| Standard | 99.06/33.36 | 93.32/72.36 | 71.07/45.10 | - | 88.62/71.63 | - |
| $\mathrm{GU}_{S^\rho_{worst}}$ | 98.78/52.48 | 93.06/72.75 | 70.68/45.19 | 60% | 87.69/71.47 | 40% |
| $\mathrm{GU}_{S^\rho_{avg}}$ | 98.78/52.48 | 93.02/72.58 | 70.67/45.35 | 60% | 87.71/**72.37** | 100% |
| $\mathrm{GU}_{\mathtt{tr(F)}}$ | 98.91/**54.12** | 93.02/**73.56** | 70.78/**45.82** | 83% | 88.40/71.86 | 60% |

Table 5, Table 6 shows the complete results for 1) training from scratch evaluating on noise-corrupted inputs, and 2) PEFT tuning for cross-lingual transfer. While all results are better than the standard training, empirically, $\mathtt{tr(F)}$ is a metric that gives a better winning rate compared to a random hyperparameter search.

Table 6: Cross-lingual transfer results of standard training and using all 3 metrics to determine the unfreezing interval $\hat{k}$ for gradual unfreezing (GU), best OOD results are bolded. WR stands for winning rate, averaged over 10 randomly sampled $k$ per training dataset.

| Method | XQuAD F1- En/X-ling | EM- En/X-ling | MLQA F1- X-ling | EM- X-ling | XNLI Avg- En/X-ling | WR |
|---|---|---|---|---|---|---|
| Standard | 82.96/68.72 | 71.39/52.64 | 56.27 | 40.93 | 83.17/71.84 | - |
| $\text{GU}_{S^\rho_{worst}}$ | 83.78/70.09 | 72.10/54.17 | 57.86 | 42.02 | 82.83/72.13 | 45% |
| $\text{GU}_{S^\rho_{avg}}$ | 83.84/70.00 | 72.12/53.69 | 58.10 | 42.03 | 83.03/72.27 | 40% |
| $\text{GU}_{\text{tr}(F)}$ | 83.77/**70.70** | 72.33/**54.40** | **58.47** | **42.31** | 83.36/**72.49** | 80% |

Table 7: Domain generalization results of standard training and using $\text{tr}(F)$ to determine unfreezing interval $\hat{k}$ for gradual unfreezing (GU), best OOD results are bolded. WR stands for winning rate, averaged over 10 randomly sampled $k$ per training dataset.

| Method | DomainNet OOD | WR |
|---|---|---|
| Standard | 35.34 | - |
| $\text{GU}_{S^\rho_{worst}}$ | **37.95** | 90% |
| $\text{GU}_{S^\rho_{avg}}$ | 37.80 | 90% |
| $\text{GU}_{\text{tr}(F)}$ | 37.86 | 90% |

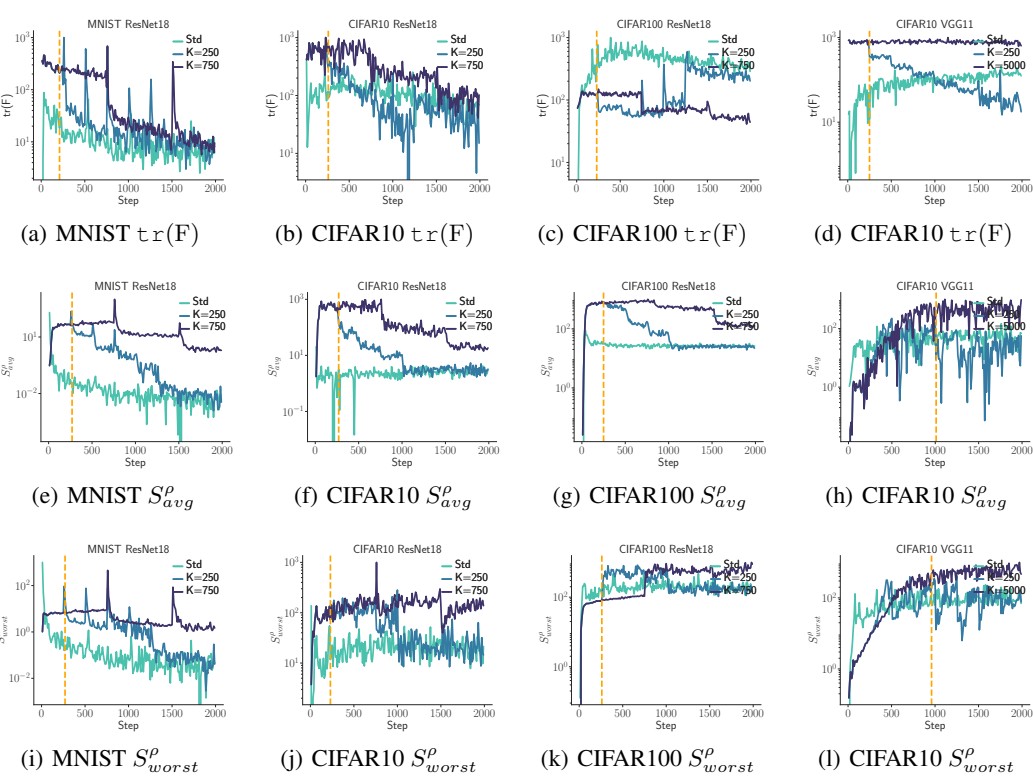

(a) MNIST $\text{tr}(F)$    (b) CIFAR10 $\text{tr}(F)$    (c) CIFAR100 $\text{tr}(F)$    (d) CIFAR10 $\text{tr}(F)$

(e) MNIST $S^\rho_{avg}$    (f) CIFAR10 $S^\rho_{avg}$    (g) CIFAR100 $S^\rho_{avg}$    (h) CIFAR10 $S^\rho_{avg}$

(i) MNIST $S^\rho_{worst}$    (j) CIFAR10 $S^\rho_{worst}$    (k) CIFAR100 $S^\rho_{worst}$    (l) CIFAR10 $S^\rho_{worst}$

Figure 6: Learning dynamics with $\hat{k}$ given by Algorithm 2 (vertical line).

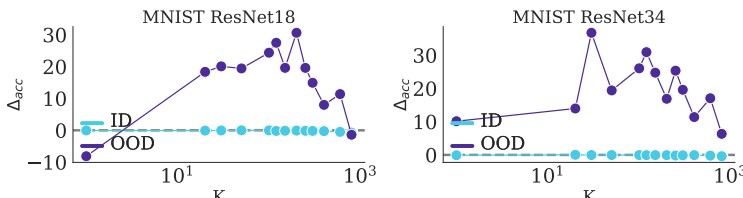

Figure 7: Changes in ID and OOD evaluation results when unfreezing parameters at different times (i.e., $k$) for ResNet18 and ResNet34. The $\Delta$ is calculated by subtracting the gradual unfreezing results from standard training, averaging 6 runs. The x-axis is on a log scale.

Table 8: ID and OOD evaluation results of ResNet18 and ResNet34 on MNIST. $k^*$ is the optimal $k$ that produces the best average OOD results.

|  | ResNet18 ID/OOD | ResNet34 ID/OOD |
| --- | --- | --- |
| Std | 99.06/33.36 | 99.24/10.04 |
| $k^*$ | 98.98/63.99 | 99.20/46.80 |

## F IMPACT AND TIME-SENSITIVITY OF INTERVENTIONS ON OUT-OF-DISTRIBUTION RESULTS

### F.1 MODEL SIZE

We further experimented with a larger ResNet (ResNet34) on the MNIST dataset (chosen for its efficiency as we needed to conduct 86 experiments to generate the subfigure). The time-sensitive nature remains consistent across ResNet models of different depths. Figure 7 shows the results and the numerical results are in Table 8, indicating that this time sensitivity in OOD generalization persists across models with different depths. Interestingly, the larger model (ResNet34) shows no significant differences in ID results while exhibiting greater variation and degradation in OOD results compared to the smaller model (ResNet18).

### F.2 LEARNING RATE WARM-UP

We also experiment with a simple learning rate warm-up (step function, single step), starting from a reduced learning rate (1/10 or 1/5 of the target learning rate) and switching at a specific time $k$. We evaluate this approach with: 1) ResNet18 trained from scratch, test on noise-corrupted CIFAR10, starting at 1/10 of the target learning rate, and 2) fine-tuning the pre-trained ViT, test on the Office-Home dataset (domain shift) from 1/5 of the target rate. The results are in Figure 8.

Although the improvements are smaller compared to gradual unfreezing, adjusting the learning rate switch timing increases OOD accuracy by up to +1.31% with ResNet18, with minimal impact on ID performance. The maximum improvements on the Office-Home dataset is +1.67%. This again highlights the importance of timing in applying or removing interventions for better OOD generalization.

### F.3 FISHER PENALTY

Prior work shows that regularizing $\text{tr}(F)$ can help with ID generalization (Jastrzebski et al., 2021) (training from scratch). Let $\mathcal{J}$ be the original loss, the total loss with Fisher penalty is in Eqn. 5. Following the simple CNN setting in (Jastrzebski et al., 2021, Appendix I.2), we train a simple 4-layer CNN (with one MaxPooling layer, no dropout) and a final fully connected layer of 128 hidden units on the CIFAR10 dataset from scratch with data augmentations. The model is trained for 300 epochs using an SGD optimizer with batch size 128, momentum 0.9, and a learning rate decay of 0.1 after epochs 150 and 225. We use a starting learning rate of 0.001 (a smaller learning rate than the default) and apply the Fisher penalty (FP) with a strength of 0.01 ($\alpha$) every 10 steps. The model is evaluated with noise-corrupted inputs (i.e., CIFAR10-C) during the test.

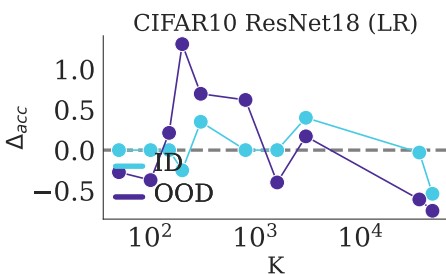
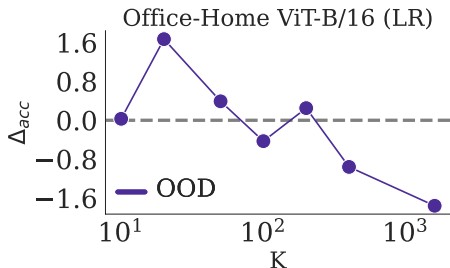

(a) The maximum improvements in OOD results (noise-corrupted inputs) are +1.31%.

(b) The maximum improvements in OOD results (domain shift) are +1.67%.

Figure 8: Changes in ID and OOD evaluation results when unfreezing parameters at different times (i.e., $k$) highlight the early training period's influence on OOD generalization in different settings with learning rate warm-up (step function, single step). The $\Delta$ is calculated by subtracting the gradual unfreezing results from standard training, averaging 4 runs. The x-axis is on a log scale.

$$\mathcal{J}_{total} = \mathcal{J} + \alpha * \mathtt{tr}(F).\tag{5}$$

First, in this small learning rate setting, the ID results improved from 84.45 to 85.39 on average over 4 random seeds (compared to no FP) when we applied FP to the training. Then, we experiment with delaying the application of the FP regularizer by $k$ steps, and Figure 9 shows the results compared to applying FP from the beginning of the training. Delaying the application of the FP decreases ID results by a small fraction, but increases OOD results compared to no delay. The best $k$ appears to be between 1000 to 3000 steps (largest OOD increase, smallest ID decrease). Figure 10 shows the learning dynamics with no FP penalty (Std) and with the FP applied with no delay ($k$=0) or a delay of 2000 steps (i.e., $k$=2000). The sharpness profile of the $k$=2000 curve follows a high-to-low trend.

The stabilization of sharpness and $\mathtt{tr}(F)$ once again coincides with the period of improved OOD results in Figure 9. This supports our hypothesis that the point at which sharpness and $\mathtt{tr}(F)$ stabilize marks the optimal time to apply a regularizer that reduces sharpness during early training.

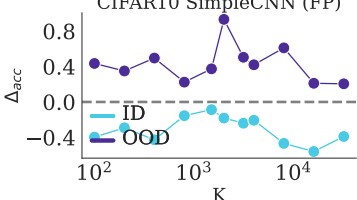

We also use the same algorithm (Appendix E) to determine the time for application of FP (i.e., find a time $k$ when the sharpness metrics or FIM stabilizes). The ID/OOD results are 85.04/66.25, 85.22/66.65 and 84.98/65.97, determined using $S^\rho_{worst}$, $S^\rho_{avg}$ and $\mathtt{tr}(F)$, respectively ($k$=1980/1930/2410). As expected, the OOD results are better than applying the FP from the beginning of the training.

Figure 9: Change in ID and OOD evaluation results when applying Fisher penalty at different times.

### F.4 SPURIOUS CORRELATION EXPERIMENT

We hypothesize that the effectiveness of gradual unfreezing on OOD generalization stems from implicitly regularizing learning from possible spurious features present in the dataset. To test this hypothesis, we experiment with a pre-trained ResNet18 (on ImageNet) and the WaterBirds dataset (Sagawa et al., 2020) which the spurious features are known. Training hyperparameters are determined based on Izmailov et al. (2022) (learning rate=3e-3, epochs=100, weight decay=1e-4, batch size=32, SGD optimizer, 4 runs).

In this experiment, the accuracy and worst-group accuracy (WGA) for standard training (Empirical Risk Minimization) were 96.88% and 56.93%, respectively, compared to 96.88% and 58.29% with gradual unfreezing. This result confirms that gradual unfreezing achieves better OOD results by implicitly regularizing spurious correlations.

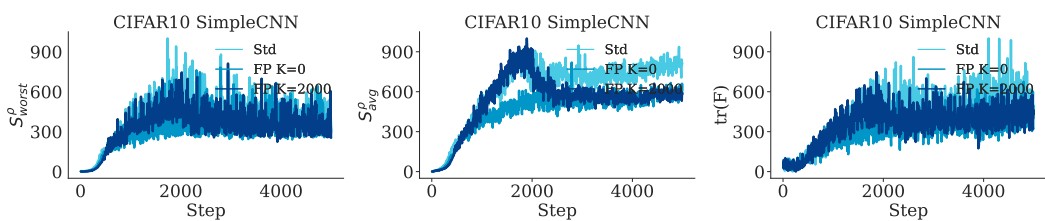

Figure 10: Learning dynamics of a simple CNN model on CIFAR10 with and without the application of the Fisher penalty from Jastrzebski et al. (2021). Std means standard training without using the Fisher penalty. $k=0$ means the Fisher penalty is applied at the start of training. $k=2000$ means the Fisher penalty is applied with a delay of 2000 steps. The y-axis is normalized and results are smoothed using a rolling window for better visualization.

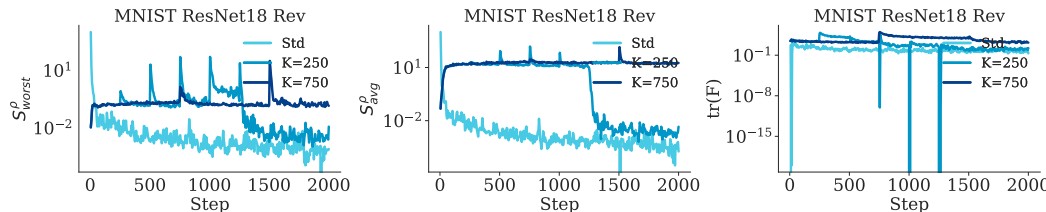

Figure 11: Learning dynamics of a ResNet 18 model trained with MNIST. *Rev* indicates that the unfreezing order progresses from the bottom layers to the top layers. Similar to the trends using top-down order, higher sharpness and $\mathtt{tr}(F)$ are observed.

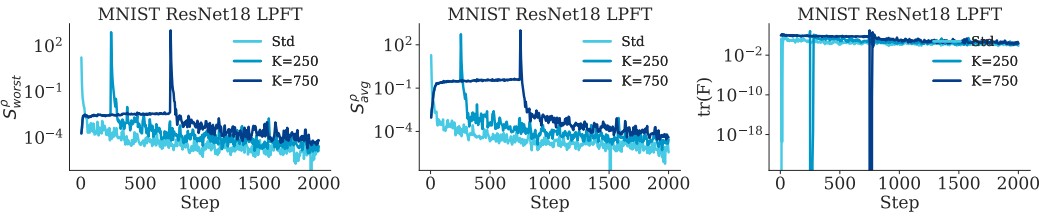

Figure 12: Learning dynamics of a ResNet 18 model trained with MNIST using LP-FT as in Kumar et al. (2022). Similar to the trends using top-down order, higher sharpness and $\mathtt{tr}(F)$ are observed.

## G    PROPERTIES OF THE FINAL SOLUTIONS AND OOD RESULTS

Changing the learning dynamics in the early period of training inevitably results in different final solutions. We plot the final solution's $\lambda_{max}$ (largest eigenvalue of training data feature), $S^\rho_{worst}$ and $S^\rho_{avg}$ against the OOD test results in Figure 13 respectively.

While in general the sharpness measures and OOD have negative correlations (i.e., the smaller the sharpness values the better, especially $S^\rho_{worst}$ has a consistent negative correlation), they are not always statistically significant (e.g., for MNIST). The learning rate has a big impact on the final solutions' sharpness. Furthermore, such as in Figure 13 (c), we can even attain slightly positive correlations. Our results complement the findings in Andriushchenko et al. (2023), which serve as evidence pointing towards the need for developing robust new metrics and thorough investigation for OOD generalization.

## H    ADDITIONAL LEARNING DYNAMICS

### H.1    GRADIENT SIMILARITY

Figure 14 illustrates additional gradient similarity (between mini-batch gradients and the full gradient, §5.2) during the early period of training for ResNet18. On average, gradient similarity increases when trainable parameters are constrained. Additionally, higher layers exhibit greater similarity to the full gradient at the beginning of training.

### H.2    FEATURE RANK

Figure 15 shows the evolution of feature ranks before the classification head for the first 2000 training steps. We observe that standard training typically starts with a lower feature rank, and as training progresses, the feature rank gradually increases. When withholding parameters from training, the feature ranks are high at the beginning of the learning period. As parameters are gradually released, the feature ranks decrease compared to their initial values.

### H.3    SQUAD

In Figure 16, we present the learning dynamics for XLM-R with SQuAD in the early period of learning. The learning dynamics show a similar trend as the SQuAD dataset, the $S^\rho_{worst}$ value is also negative, and withholding trainable parameters increases the $S^\rho_{worst}$ during training based on Eqn. 4 in our main paper.

### H.4    TRAINING FROM SCRATCH

Figure 17 shows all the learning dynamics in the early period of training using the same learning rate in §D with gradual unfreezing. Figure 18 shows the learning dynamics in the early period of training using 1/10th of the learning rate specified in §D with gradual unfreezing. We observe consistent trends.

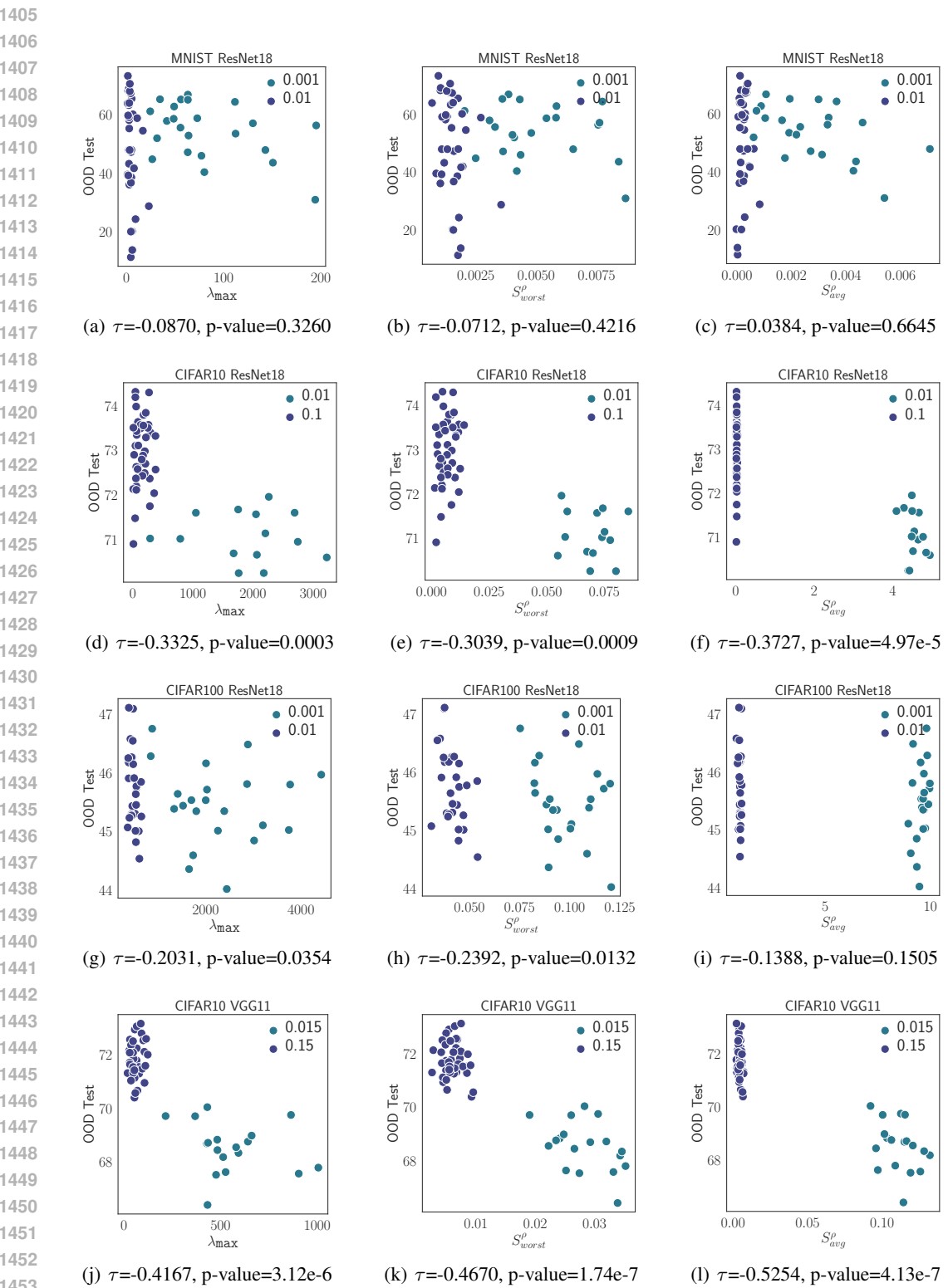

Figure 13: Final feature $\lambda_{max}$, $S^{\rho}_{worst}$, and $S^{\rho}_{avg}$ versus the OOD test results (coloured by learning rate), labelled with Kendall's $\tau$ and p-value.

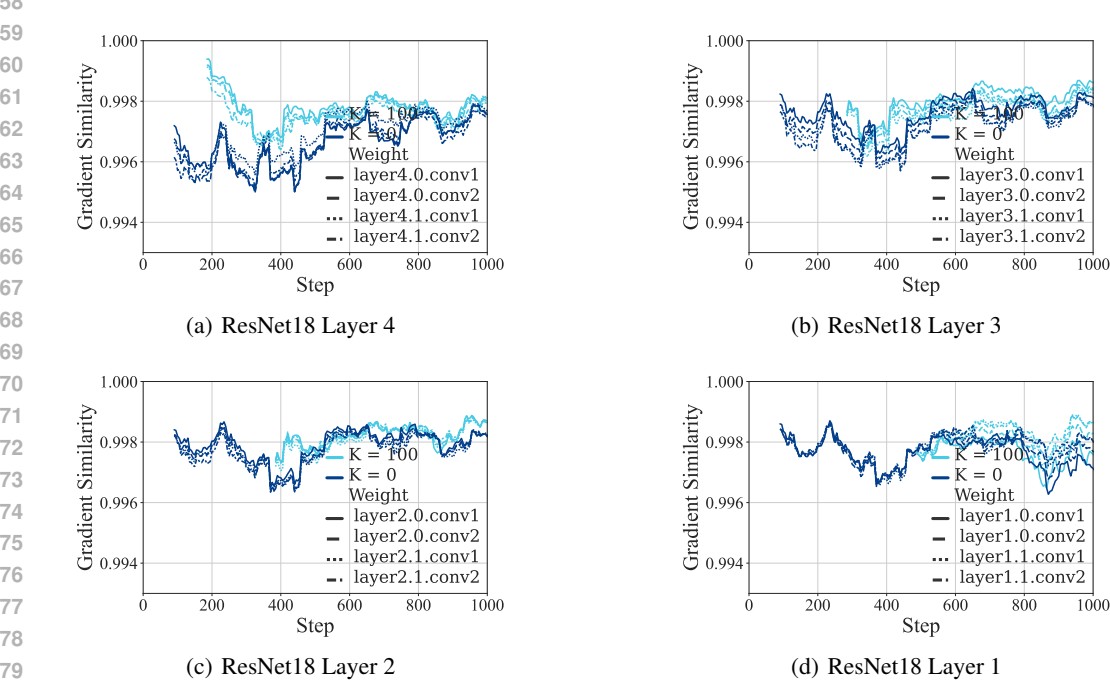

Figure 14: Gradient similarity (mini-batch vs full-data) for the convolution layers in ResNet18 during training with CIFAR10. The mini-batch gradient is, on average, more similar to the full-data gradient when gradual unfreezing is applied (K=100) compared to standard training (K=0). This effect is more pronounced in the higher layers.

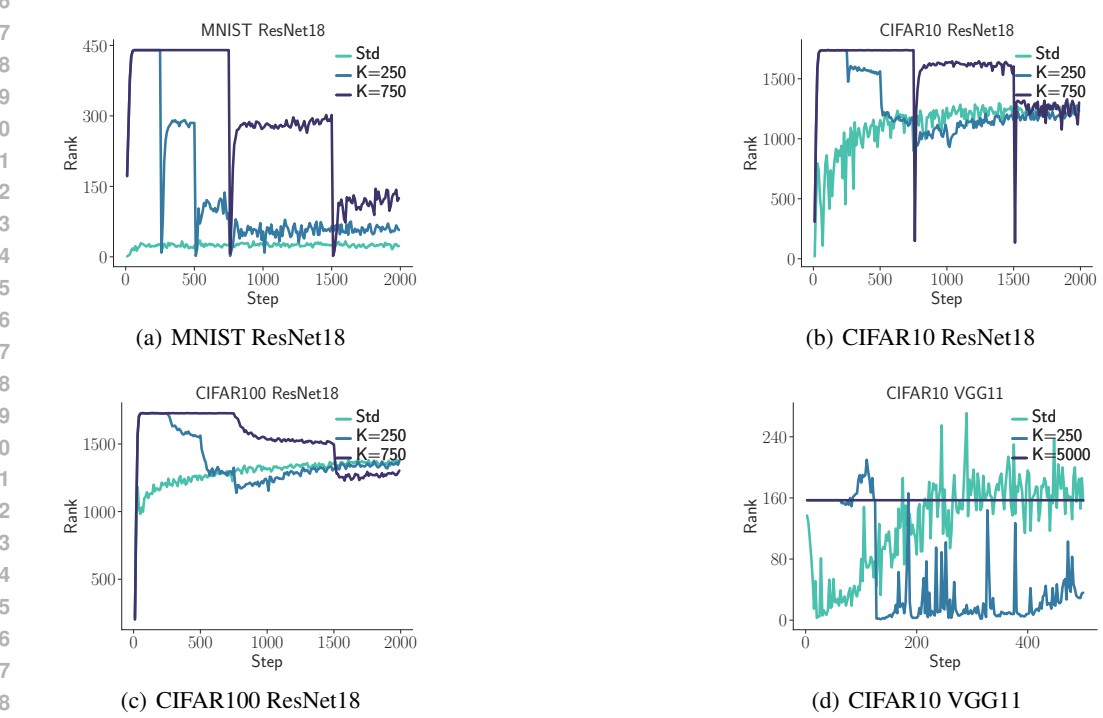

Figure 15: Change of feature ranks before the classification head. The sudden decrease in feature ranks is due to unfreezing the trainable parameters.

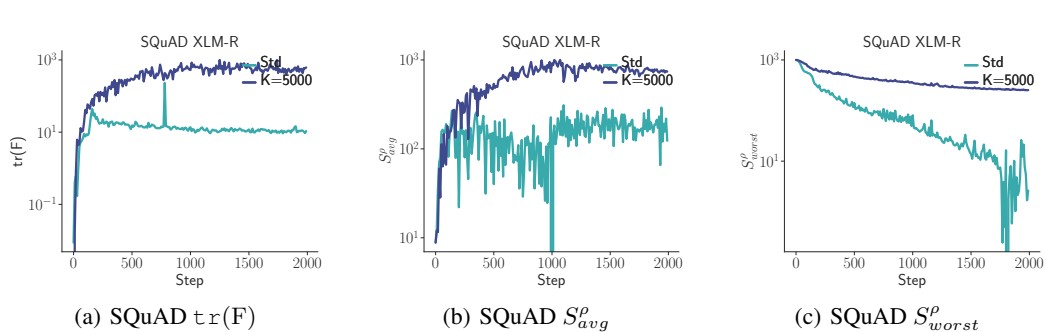

(a) SQuAD $\mathtt{tr}(\mathrm{F})$  (b) SQuAD $S_{avg}^{\rho}$  (c) SQuAD $S_{worst}^{\rho}$

Figure 16: Learning dynamics of XLM-R with LoRA training with SQuAD, y-axis for figures are in the log scale, the original value sharpness value in sub-figure (c) is negative where we take the absolute value before visualization. All values are normalized between 0 and 1000 for visualization.

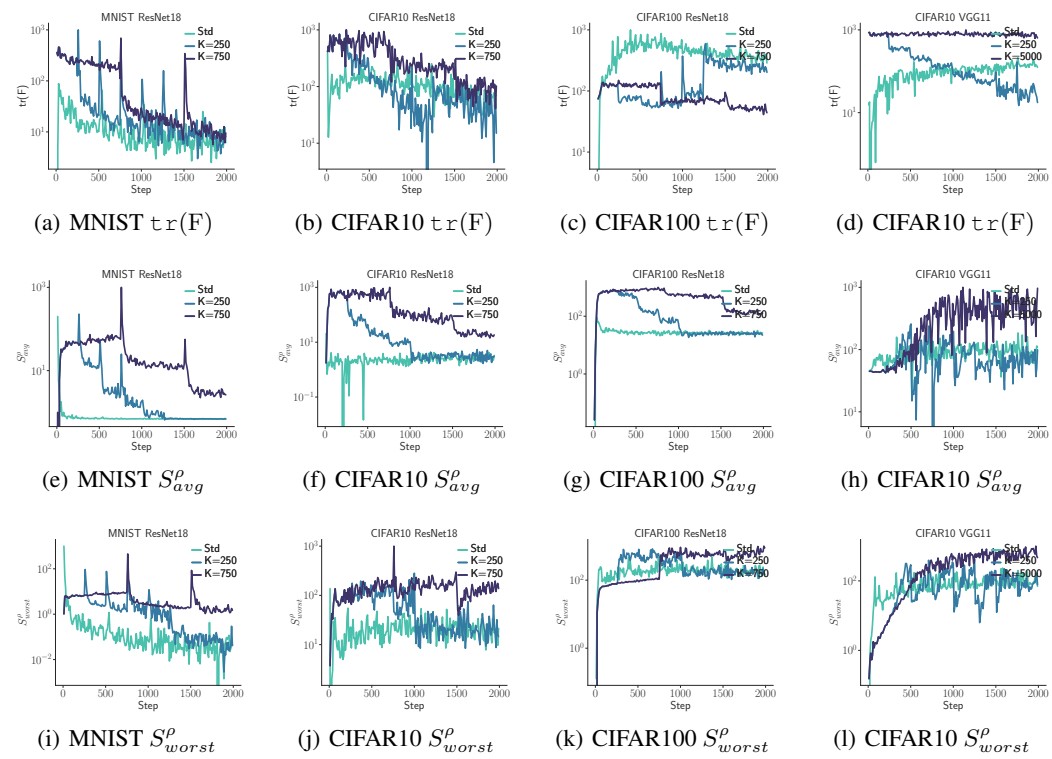

(a) MNIST $\mathtt{tr}(\mathrm{F})$  (b) CIFAR10 $\mathtt{tr}(\mathrm{F})$  (c) CIFAR100 $\mathtt{tr}(\mathrm{F})$  (d) CIFAR10 $\mathtt{tr}(\mathrm{F})$

(e) MNIST $S_{avg}^{\rho}$  (f) CIFAR10 $S_{avg}^{\rho}$  (g) CIFAR100 $S_{avg}^{\rho}$  (h) CIFAR10 $S_{avg}^{\rho}$

(i) MNIST $S_{worst}^{\rho}$  (j) CIFAR10 $S_{worst}^{\rho}$  (k) CIFAR100 $S_{worst}^{\rho}$  (l) CIFAR10 $S_{worst}^{\rho}$

Figure 17: Unfreezing parameters at different times affects the learning dynamics in the early period of training (with $lr_d$). We show $\mathtt{tr}(\mathrm{F})$, $S_{avg}^{\rho}$ and $S_{worst}^{\rho}$ when parameters are unfrozen at steps $k = \{250, 750\}$ for ResNet and $k = \{250, 5000\}$ for VGG, versus standard training. The y-axis uses a log scale and is normalized between 0 and 1000 for visualization.

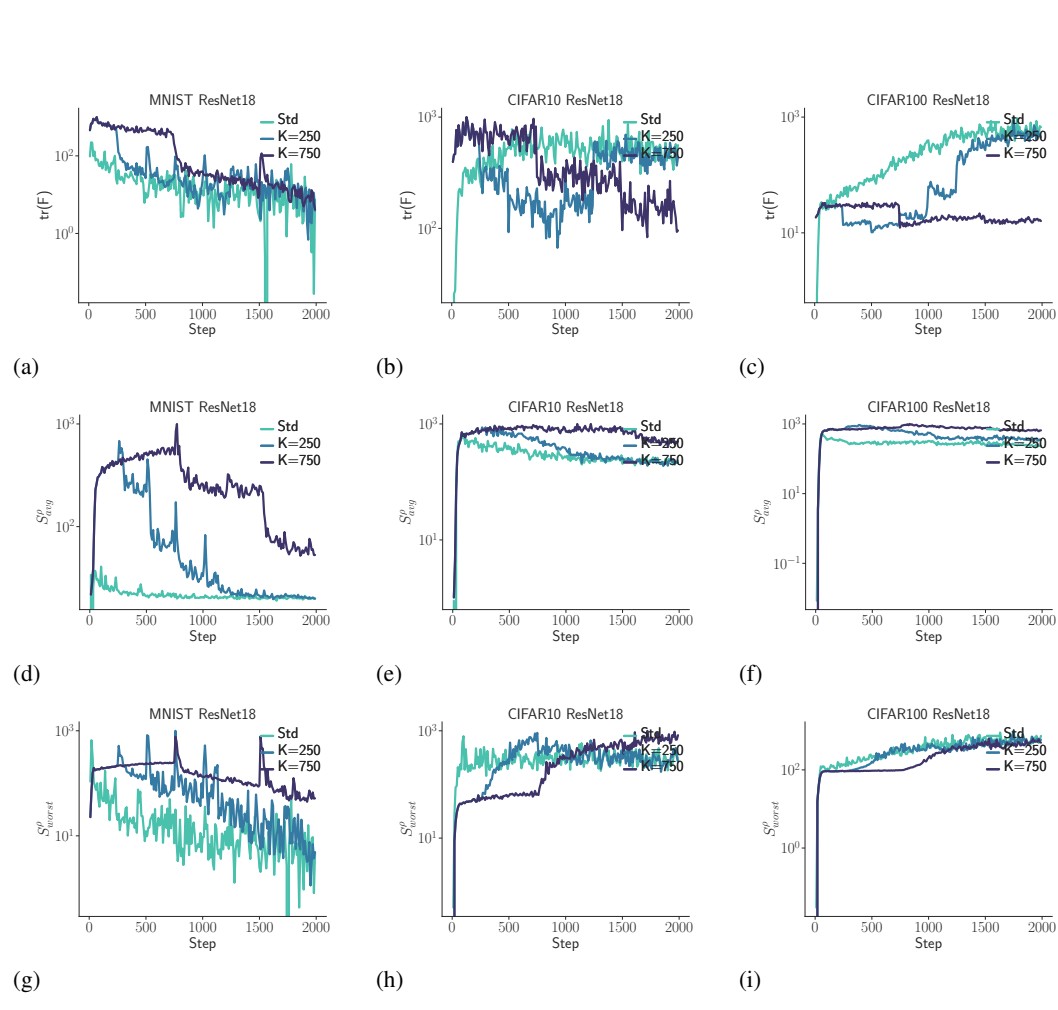

Figure 18: Unfreezing parameters at different times affect the learning dynamics in the early period of training. We show $\mathtt{tr}(\mathrm{F})$, $S^{\rho}_{avg}$ and $S^{\rho}_{worst}$ when parameters are unfrozen at steps $k = \{250, 750\}$ for ResNet, versus standard training. The y-axis uses a log scale and is normalized between 0 and 1000 for visualization. We use 1/10th of the learning rate specified in §D.

