# OpenReview forum: "Early Period of Training Impacts Adaptation for Out-of-Distribution Generalization: An Empirical Study"
_ICLR.cc/2025/Conference — Submitted to ICLR 2025_

### Official Review · Reviewer_4W2j · 2024-10-21

**Soundness:** 2
**Presentation:** 3
**Contribution:** 3
**Rating:** 6
**Confidence:** 3

**Summary:**

This paper investigates the connection between early training dynamics and out-of-distribution (OOD) generalization. By using the trace of Fisher information and sharpness as indicators to examine the effects of interventions, the authors propose a gradual unfreezing strategy that enhances OOD generalization during the early training phase without impacting in-distribution (ID) performance.

**Strengths:**

1. This work is the first to highlight the impact of the early training period on OOD generalization, offering a novel perspective.

2. The use of relative values of Fisher information and sharpness as metrics for OOD generalization is interesting.

3. The gradual unfreezing method is tested across a wide range of transfer learning tasks and backbone networks, demonstrating its general applicability.

**Weaknesses:**

1. The paper lacks comparisons to relevant baselines in some transfer learning tasks, such as Office-Home and DomainNet. Many studies in domain generalization (DG) or OOD research focus on these datasets, but the absolute improvements in OOD accuracy presented here seem modest compared to existing methods.

2. Previous work, such as [1], has shown that freezing parameters improves OOD performance. It is unclear whether the gradual unfreezing method can be considered a variation or a weaker form of such approaches. The discussion in Section 5.2 touches on this but remains somewhat unclear.

3. From a causal standpoint, it is difficult to establish a causal relationship between sharpness and OOD performance rather than mere correlation. It is possible that gradual unfreezing simultaneously affects both sharpness and OOD performance. Causal intervention strategies might be needed to investigate this in more detail, for example, by ensuring sharpness remains constant during optimization after freezing and observing the potential drop in OOD performance.

[1] Kumar A, Raghunathan A, Jones R, et al. Fine-Tuning can Distort Pretrained Features and Underperform Out-of-Distribution[C]//International Conference on Learning Representations. 2022.

**Questions:**

1. Could additional baselines from existing OOD algorithms be included to better evaluate the impact of the proposed method?

2. How does the gradual unfreezing strategy compare to other parameter-freezing approaches during early training? Could the similarities and differences be further clarified?

3. Have similar changes in Fisher information or sharpness been observed in other OOD algorithms, and if so, how do they relate to the findings in this paper?

---

> ### Author Response · Authors · 2024-11-22
> **Reviewer 4W2j**
>
> Thank you for your time and constructive feedback.
>
> W1/Q1: Comparisons with Baselines
> In this paper, our goal was to focus on evaluating the effect of gradual unfreezing (as a tool) specifically within early training dynamics, rather than proposing a SOTA method. We selected gradual unfreezing as an intervention method due to prior studies on its effectiveness in achieving SOTA in various OOD settings.
>
> W2: Gradual unfreezing shares similarities with the approach in [1] for transfer learning with a pre-trained backbone, as both involve selectively tuning parameters over time to enhance OOD performance. Gradual unfreezing can be seen as a variant of the approach in [1], offering more fine-grained algorithmic control by tuning layer-by-layer instead of focusing solely on the classification head and backbone. In Section 5.2, we demonstrate that gradual unfreezing aligns gradient similarity across batches during early training, particularly in the classification head, further supporting this connection.
>
> W3: Causal Interpretation of Sharpness and OOD Performance
> Thank you for this interesting suggestion. While our experiments establish a correlation, causality would indeed require interventions to control sharpness independently of other factors. Optimization processes (back-propagation / parameter updates), inherently affect sharpness/Fisher Information regardless of freezing or unfreezing. Currently, we lack an effective method to isolate and control these changes, which we acknowledge as a limitation and included in our limitations section (L918). We have some additional experiments in Appendix F.2 and F.3, where sharpness is regularized midway through training, yet also yield better test OOD results, further suggest that early sharpness/Fisher Information (FI) measurements may be noisy.
>
> Q2:How does the gradual unfreezing strategy compare to other parameter-freezing approaches during early training? Could the similarities and differences be further clarified?
> We used MNIST with ResNet18 training from scratch and we visualized the training dynamics for LPFT (head first, then unfreeze the layers in the backbone all together) [1] and reverse unfreezing (bottom-up). Using LPFT and with a reverse selection of layers to unfreeze during training after the unfreezing of classification head. The training dynamics are now included in the revised draft Figure 11 and Figure 12, in the appendix.
>
> Freezing parameters during the early stages of training generally increases sharpness compared to standard training. When parameters are unfrozen, it introduces fluctuations to the training dynamics but rapidly reduces sharpness.
> These findings suggest that the absolute value of sharpness/Fisher Information (FI) in the early training phase is not a reliable indicator of generalization when accounting for trainable parameters. (Note: This does not contradict findings on sharpness at convergence, as our focus is solely on the early training phase.) Instead, changes in training dynamics (sharpness/FI) is more likely to be signals of a “phase change” [2] in the learning process rather than directly predicting OOD generalization when factoring in the trainable parameters (hence, leads to our experiments with the heuristic algorithm).
>
> Q3: To the best of our knowledge, no prior work has specifically focused on this aspect of OOD behaviour.
> In our Appendix (F.3), we also found when keeping the trainable parameters constant, by applying a regularizer that reduces the FI [3] also has a similar effect. While we have not yet experimented with other conventional OOD methods, we aim to explore them in the future. As previously mentioned, we observed that the Fisher Penalty also demonstrates similar behaviour, where minimizing sharpness from the beginning does not improve OOD performance.
>
> ------
> Reference:
>
> [1] Kumar, Ananya et al. “Fine-Tuning can Distort Pretrained Features and Underperform Out-of-Distribution.” International Conference on Learning Representations (2022).
>
> [2]  Achille, Alessandro et al. “Critical Learning Periods in Deep Networks.” International Conference on Learning Representations (2018).
>
> [3] Jastrzebski, Stanislaw et al. “Catastrophic Fisher Explosion: Early Phase Fisher Matrix Impacts Generalization.” International Conference on Machine Learning (2020).

---

> > ### Comment · Reviewer_4W2j · 2024-11-23
> > **Thanks for Reply**
> >
> > Thank you for your response. I generally acknowledge the contributions of this paper, but considering that it is purely an empirical study, its significance is somewhat limited. Therefore, I will maintain my current score.

---

> > > ### Author Response · Authors · 2024-11-30
> > > **Re**
> > >
> > > Thank you for your response. We acknowledge that the empirical nature of our work is a limitation and we will like to explore the theoretical aspects in the future. Please let us know if you require any further clarification.

---

### Official Review · Reviewer_guvH · 2024-10-29

**Soundness:** 3
**Presentation:** 3
**Contribution:** 2
**Rating:** 5
**Confidence:** 4

**Summary:**

This paper studies the early learning dynamics of OOD generalization under covariate shift. The authors leveraged gradual unfreezing (i.e. gradually unfreeze blocks of parameters during training) as a tool and observed that OOD generalization can be improved if gradual unfreezing is applied at the beginning of training. In addition, they also observe a general trade-off between ID and OOD performance when varying the time interval of unfreezing. They further use Fisher information and loss sharpness as proxies to design heuristics that allow for capturing the optimal time interval of gradual unfreezing.

**Strengths:**

1. The paper is clear and easy to follow.
2. The experiments span various datasets and training setups.
3. The study of covariate shift early training dynamics, to the reviewer's knowledge, is new.

**Weaknesses:**

1. Lack of explanation on why the approach should work: while the reviewer acknowledges that (from its title) this paper is an empirical study, they found it hard to evaluate the value of findings in the paper without more explanation or intuition, especially the author proposes a heuristics-driven approach to improve OOD.
    1. explanation of gradual unfreezing: while the author provided intuition on page 6 about why gradual unfreezing would work, the reviewer found it not convincing: most of the experiments in Kumar et al. [1], if the reviewer remembers correctly, train linear probe then fine-tuning till convergence, but this paper considers early-training with small amount of training steps. In addition, Kumar et al. [1] consider transferring from a pre-trained model (that’s why the feature can be distorted). In contrast, in this paper, the authors train from scratch for part of the experiments. It is hard to imagine the benefits of first aligning the classification head (or top layers) while keeping the bottom layers (feature extractor) randomly fixed.
    2. On phases of the change of the metrics: Besides, the dynamics of metrics in Figure 5 are very different, where in some cases they are decreasing but in other cases they are increasing.
2. Related works: the reviewer found it slightly over-claimed to say that ‘the impact of the early learning period on OOD generalization remains unexplored’ throughout the paper. How the (early) learning dynamics influence the OOD generalization has been widely explored in spurious correlation, which is another major OOD generalization problem. It would be better if the authors could discuss the connection to these related works and rephrase their claims in the paper. Some examples are [2][3].
3. Robustness of the proposed heuristics: how the proposed heuristics would be able to transfer to real-world dataset is unknown. While the authors conduct experiments on multiple datasets, it is still hard to evaluate because of the lack of explanations of some components in this work (see 1.)
4. Ablation study: while we see that in Figures 2 and 3 there exists a sweet spot where ID performance remains stable while OOD performance improves, all the runs use gradual unfreezing at the beginning. A helpful ablation study shows that gradual unfreezing at the middle period or ending period of training does not exhibit this observation. That is, starting training the whole network for some epochs and then freeze all parameters and gradually unfreeze.

[1] Fine-Tuning can Distort Pretrained Features and Underperform Out-of-Distribution. Kumar et al. ICLR 2022
[2] Identifying Spurious Biases Early in Training through the Lens of Simplicity Bias. Yang et al. AISTATS 2024
[3] Complexity Matters: Feature Learning in the Presence of Spurious Correlations. Qiu et al. ICML 2024

**Questions:**

Please see the above ‘weakness’ section and provide clarification if needed.

Other questions are:

1. In Figure 5, why the k used in different subfigures different?
2. Figure 5 again, it seems that the pattern of ‘rapid change followed by stablization phase’ does not hold on the first plot of subfigure (b)(c).

Minor:

1. For a first-time reader, it is not clear what is the ‘removal of interventions’ at row 24 of the abstract, maybe consider just using ‘gradual unfreezing’ instead.

---

> ### Author Response · Authors · 2024-11-22
> **Re Reviewer guvH**
>
> We would like to thank reviewers for their time and feedback.
>
> W1: We recognize the importance of providing clearer intuition on why gradual unfreezing supports OOD generalization. In our experiments, we explored three different settings: one involving training from scratch and two involving fine-tuning with pre-learned features. We agree that the two fine-tuning settings are comparable to the approach of Kumar et al. [1]. Restricting trainable parameters acts as a form of regularization, promoting better gradient alignment and increased robustness to spurious correlations. We think this is the underlying mechanism at play when training from scratch (section 5.2).
>
> W2: We appreciate your insightful suggestion on the additional work on spurious correlations. Our empirical evidence seems to suggest this could be related, especially when existing covariate shifts evaluation dataset don’t control for spurious correlations. We have updated our related work section (L108) to include the suggested recent work on early training and spurious correlation.
>
> W3: We acknowledge the limitation in generalizing this heuristic and intend to expand our study in future work to better validate robustness across real-world scenarios. Our goal is to use heuristics to validate our hypothesis, and we believe our experiments achieve this objective. We really appreciate the reviewer's suggestion and include related discussion in our limitations section.
>
> W4: Ablation Study on Timing of Gradual Unfreezing
> Thank you for this insightful suggestion. We performed additional experiments where gradual unfreezing was delayed to mid- and late-training phases. We first allow full parameter training, then during the mid- or late- training, freeze all the parameters, and perform unfreezing (standard training → freezing all parameters → gradual unfreezing).  Here we report the results using MNIST when training from scratch (using the same training hyperparameters specified in our current draft appendix, hyperparameter search 3 values of the unfreezing interval k, results averaged over 6 random seeds).
>
> The best ID/OOD results are as of following:
>   - Freezing-Unfreezing starts at 500 steps, best results: 99.18 / 32.29
>   - Freezing-Unfreezing starts at 1000 steps, best results: 99.22 / 30.71
>   - Freezing-Unfreezing starts at 2000 steps, best results: 99.26 / 39.12
>
> For comparison, the early period unfreezing gives 98.98 / 63.99.
>
> Q1/Q2 - Clarifications on Figure 5
> The values of k adapted in Figure 5 are randomly selected to help with visualisation. Regarding the “rapid change followed by stabilization phase,” our observations indicate that, depending on the dataset and initialization, metrics like tr(F) and sharpness can stabilise at varying times. This is evident in the differences seen across Figure 5 subplots, emphasising that heuristic adjustments may be necessary based on training dynamics.
>
> Q3: We appreciate the suggestion to replace "removal of interventions" with "gradual unfreezing" for clarity. We revised this to ensure the terminology is consistent and accessible for first-time readers.

---

> > ### Comment · Reviewer_guvH · 2024-11-25
> >
> > I thank the authors for their rebuttal.
> > Regarding W4, the results are interesting and helpful, I suggest adding it (maybe with more datasets) in the next version of the draft.
> > I still have some concerns regarding W1 (which was my major concern).
> > - When training from scratch (the analysis in Kumar et al. does not apply), why gradual unfreezing in early training is so important for OOD performance (as shown with the author's new ablation study)? I agree with Reviewer pcei that the paper lacks 'understanding' on this front but it claims so in its current format (e.g. L028). While the authors made hypotheses on gradient similarity (GS) and spurious features, I think the difference between standard training and gradual unfreezing in both cases are marginal (i.e. the GS is always very high in Figure 4 and OOD performance in Appendix F.4 improves marginally as in a binary classification problem). Besides, is it correct that if gradient similarity is the cause, then full-batch training would give an improved OOD generalization?
> >
> > For the above reason, I had to maintain my current score but I am open to discussion and will await to see other reviewers responses.

---

> > > ### Author Response · Authors · 2024-11-30
> > > **Re**
> > >
> > > Thank you very much for the discussion.
> > >
> > > We will revise L028 to be more rigorous with our writing.
> > >
> > > Regarding whether full-batch gradients help with OOD: We have not explored this ourselves. However, it is an intriguing suggestion. Based on our current understanding, we hypothesize that it could be effective during the early stages of training (e.g., using full-batch gradients initially, followed by mini-batch gradients). Nonetheless, this hypothesis requires thorough testing in future work.

---

> > > > ### Author Response · Authors · 2024-12-04
> > > > **Re guvH**
> > > >
> > > > Dear Reviewer,
> > > > Thank you very much for the valuable discussions during the rebuttal period. If we have adequately addressed your concerns, we kindly request your consideration in revising the score.
> > > >
> > > > We truly appreciate your time and effort, thank you again.

---

### Official Review · Reviewer_pcei · 2024-11-01

**Soundness:** 2
**Presentation:** 1
**Contribution:** 2
**Rating:** 3
**Confidence:** 4

**Summary:**

The authors investigate whether gradual increase of the parameters in a neural network during early training helps with boosting OOD performance. To this end, the authors propose simple strategy of gradually training more parameters in a network, starting with the earlier layers and blocks until the full model is trained. This introduces a parameter, $k$, which determines the #steps until the next block in unfrozen. ID and OOD performance is tracked with this strategy on a variety of language and vision datasets under different distribution shifts. With an optimal $k$, the authors show boost in OOD performance. In the second half of the paper, the authors discuss how $k$ can be optimally chosen based on Fisher Information or sharpness, and results are presented under this heuristic.

**Strengths:**

The paper is generally easy to follow. The problem setup is clear, the proposed method conceptually simple to implement. A variety of datasets are included in the study. The results show possibilities to increased OOD performance on these datasets with the right gradual unfreezing schedule.

**Weaknesses:**

### Summary

The paper has two big issues:

First, the flow of writing: The first half of the paper addresses a possibility to increase the OOD performance depending on optimal selection of the unfreezing schedule, with little control experiments. These results appear strong. The second half of the paper then discusses how to select the unfreezing schedule, which sets the results much more in perspective. A better way, in my opinions, would be to upfront show the final method and its empirical performance: what is the proposed selection scheme for $k$, and what is the resulting performance on ID and OOD test cases across dataset and models?

Second, missing controls and baselines. The paper claims to contribute to the “understanding” of early training dynamics, but really only shows one special case of addressing this problem. The authors claim previous methods focus on ID performance and failed to address OOD improvements, but none of the methods were implemented and ran as baselines. The unfreezing schedule is also not well motivated and arbitrarily presented. What about randomly unfreezing? What about doing this not per block, but truly randomly across network parameters? What about unfreezing top to bottom? How do the methods stack up against established strategies of

I feel like addressing any subset of these questions would greatly enhance readability, quality of investigation, and impact of the paper. Please find an additional list of weaknesses to address below (if possible, please ref their labels W1,... etc) during the rebuttal:

### Major Weaknesses

**W1.** The results presented in Figure 2 and 3 are interesting, but not sufficient to back up the authors’ claims about improvement of OOD performance. Namely, all metrics are reported directly on the test sets, which in practice are not observable, and this is not appropriately discussed in the paper. It is unclear how the optimal k in Fig. 2 and 3 would be chosen. In a typical experiments, it would be selected based on ID validation performance, and for quite a few of the presented settings, this would result in much smaller gains (or even decrease in performance).

**W2.** it is good that experiments were run across multiple seeds, but the authors should reflect these in the plots and update all plots with appropriate error bars (e.g. SEM or 95% CIs), e.g. in Fig. 2-4, and also in Fig. 5. In the tables, the WR should also be equipped with error bars.

**W3.** Is it actually required to unfreeze the model step by step? There are little controls against this proposal. What happens if blocks are unfrozen randomly? Or from last to first layer, instead vice-versa? More control experiments here would strengthen the claims made.

**W4.** The authors cite other strategies for adapting early changes to network training for improved ID performance. How do these strategies stack up against the proposed method for OOD improvements? Adding some of these comparisons would strengthen the proposed method.

**W5.** Section 3.1 is not well written. Not all variables and symbols are defined (e.g. P_w in Eq. (1)), and some of the sentences are broken. There are also multiple statements about the Fisher matrix, like “A larger Fisher information…”, “tr(F) correlates well with the full Fisher information…” which are all imprecise given that F is a matrix. I would suggest to rewrite 3.1 to improve clarity.

**W6.** The study is limited in the number of models that are investigated. It would corroborate the paper story if the findings could be demonstrated for a larger set of model architectures.

**W7.** Significance: The performance improvements in Table 1 are really slim, except for MNIST. On CIFAR, we get 72.36 vs. 73.56 and 45.10 vs. 45.82%. It would really help to contextualise this against other ways of making model training more robust. What happens when models are trained with robust pre-training techniques? Here, improvements are typically much bigger than 1-2% points.

**Questions:**

**Q1** In the conclusion, you claim that the paper contributes to a deeper understanding of the early period of training. Could you clarify what this understanding entails? It is unclear from how I read the paper. What does the empirical performance tells us about what happens during the early training phase within the network?

**Q2** After designing the final method, it would be interesting to apply it to larger scale, real-world applications cases (without much tuning). For instance, how does the method perform for increasing robustness on a dataset like ImageNet-C or ImageNet-R?

**Q3** The title references “adaptation”, but this is not part of the investigation. Could you clarify?

---

> ### Author Response · Authors · 2024-11-22
> **Re Reviewer pcei**
>
> We thank the reviewer for their time and suggestions.
>
> We would like to clarify a misunderstanding. As stated in our original submission, "only ID data is used for model selection (**L199**)," which aligns with the reviewer’s suggestion. All training dynamic metrics were calculated using training data, as described in **L952**.
>
> We appreciate the reviewer’s comments on the necessity of unfreezing layers sequentially versus randomly. To clarify, top-down unfreezing is not proposed as a novel method but rather as an analytical tool for our empirical study (we add discussion to this in Sec 3.3 in the revision). Prior work [1-5] has demonstrated that top-down unfreezing performs best in practical OOD generalization scenarios. Specifically, [2] and [3] compared top-down unfreezing with reverse schedules (bottom-up) and alternative selection strategies (e.g., random or Fisher-based), finding that these approaches performed worse or comparably. Based on this evidence, we chose to focus on the top-down schedule for this study.
>
> Additionally, we present results for MNIST training from scratch using the same setup, where the best-performing bottom-up averages for ID/OOD across six runs are 99.17 / 44.73, respectively. For reference, the best-performing top-down performances are 98.98 / 63.99.
>
> We have made a significant effort to study a diverse range of models, architectures, and domains, including ResNet18, VGG11, SimpleCNN, pretrained Vision Transformers, and XLM-RoBERTa, across both training from scratch and fine-tuning settings aligning with the models used in prior studies. Despite limited academic compute resources, we conducted a large number of experiments to produce our empirical results, *over 300 experiments for Figure 2 alone*. We believe our current results provide robust and sufficient evidence to support our conclusions.
>
> We emphasize that the observed correlation between OOD performance changes and the timing of intervention removal is both novel and important. Additionally, our training dynamic analysis uncovers a compelling pattern: early metrics such as sharpness or Fisher Information (FI) do not reliably indicate OOD generalization. This is particularly noteworthy, as prior studies often assume trainable parameters are constant or that all parameters are trainable, which ignores the efficient tuning approaches in current transfer learning settings.
>
> Finally, we would like to emphasize that the primary goal of this work is to demonstrate the existence of a critical period in OOD generalization and analyze its impact. The focus is not on proposing a novel method to improve OOD generalization but rather on gaining an empirical understanding of this critical period. Therefore, the magnitude of performance improvement is not the central concern of this study.
>
> We hope these clarifications address your concerns and further highlight the contributions of our study. Thank you for your constructive feedback.
>
>
> -------
> References:
>
> [1] Kumar et al. “Fine-Tuning can Distort Pretrained Features and Underperform Out-of-Distribution.” International Conference on Learning Representations (2022).
>
> [2] Lee et al. “Surgical Fine-Tuning Improves Adaptation to Distribution Shifts.” International Conference on Learning Representations (2023).
>
> [3] Liu et al.. “FUN with Fisher: Improving Generalization of Adapter-Based Cross-lingual Transfer with Scheduled Unfreezing.” North American Chapter of the Association for Computational Linguistics (2023).
>
> [4] Howard and Ruder. “Universal Language Model Fine-tuning for Text Classification.” Annual Meeting of the Association for Computational Linguistics (2018).
>
> [5] Reinhardt et al. “Improving Vision-Language Cross-Lingual Transfer with Scheduled Unfreezing.” Proceedings of the 3rd Workshop on Advances in Language and Vision Research (ALVR) (2024).

---

> > ### Comment · Reviewer_pcei · 2024-11-25
> >
> > Thanks a lot for the reply.
> >
> > Side note: I have a bit of a hard time mapping the rebuttal onto the various comments I made. As a general comment, as I outlined in my review, it would help a lot if you could reference the weakness you address in your reply (maybe actually edit your original review to make this clearer, if openreview allows that) --- the AC will surely appreciate this as well.
> >
> > Besides this, I think the paper was not yet updated --- are you still planning to do this before the end of the discussion phase? Especially the addition of error bars would help judging the results better in Fig. 2/3. The methods are also not updated.
> >
> > > As stated in our original submission, "only ID data is used for model selection (L199)," which aligns with the reviewer’s suggestion.
> >
> > I do not see this being true in Figure 2. You report a "maximum OOD accuracy improvement", which is clearly selected by using the OOD, not the ID data. This causes a discrepancy: In Figure 2, for CIFAR, you report 3.25% points improvement. In Table 1, with the heuristic selection strategy, you report 72.36±0.63 vs. 73.56±0.45, i.e. a 1.23% point improvement.
> >
> > I think it is very important to not draw a misleading picture here. The section is about "out of distribution generalization". This implies that you handle the OOD dataset as if you did not knew about it. Which means, there is no way to pick the correct parameter $k$ in Figure 2 and Figure 3 to claim the reported boosts.
> >
> > There are various ways to fix this and put into perspective, but the paper def. needs a revision in this regard. A minimum variant of this is to clearly state in ll. 319-323 that the reported gains in these section are merely to inform later selection strategies, and are not real improvements in OOD performance from a methods perspective. Another step should include pulling the limitations sections in Appendix A into the main paper.
> >
> > There is additional circularity: The metric plots in Fig 5 explore both the metrics, but also consider different interventions based on k. Can you comment on this?
> >
> > While thinking about this more, I got the following questions, which would be good to clarify before proceeding further:
> >
> > - "after the initial rapid change of sharpness": Where is this rapid change shown and quantified?
> > - "Criterion 2) is evident across all figures in our prior experiments in §5, as larger values of k consistently degrade both ID and OOD performance": is $kL$ in Algorithm 1 always smaller than $N$ for all experiments?
> > - Appendix B mentioned three models, but Table 3 mentions 4, including VIT.
> > - How large is the ViT, 12 layers? Then for Office, you report 5k steps, does this mean the model is never fully trained? For DomainNet you report 15k steps. How does this step size related to k?
> > - For MNIST, 10 epochs with 60k samples / 128 samples per batch = 4.6k steps, is that correct? Which would again mean that the ResNet18 is not really unfrozen with a large parameter k?
> > - Just to be sure, in Figure 2, you actually show the unfreezing schedule parameter $k$, and not the steps, correct?
> >
> >
> > minor
> >
> > - Figure 5 caption, "worst"
> > - In the Algorithm 1, $N$ is not defined

---

> ### Author Response · Authors · 2024-11-30
> **Re pcei**
>
> Thank you very much for the discussion.
>
> - We will include an error bar to all our figures in a future revision.
>
> - Checkpoint selection & Figure 2: To clarify, for a given K value, a checkpoint is selected based on ID results, and the corresponding OOD results are plotted for analysis. This approach demonstrates the early training effect shown in Figure 2, as evaluating on OOD data is essential to observe this behaviour without an alternative. We are happy to revise the "maximum OOD accuracy improvement" to “maximum **possible** OOD accuracy improvement” to emphasize on this in a revision.
>
> - Figure 2 vs Table 1: As the reviewer suggested, the best possible improvements in Figure 2 are achieved by looking at OOD data. In contrast, in Table 1 the K is determined by the heuristic algorithm purely on training data learning dynamics only. This shows the potential of achieving such maximum possible OOD improvements in a realistic setting (i.e. *no* access to OOD data during hyperparameter selection).
>
> - Thank you very much for the additional suggestions on writing and pulling limitation sections into the main paper. We will incorporate the suggestions in a revision.
>
> - Fig. 5: Only GU is used in Figure 5, with different K values. Std is the vanilla standard training (no interventions).
>
> To answer your additional questions:
>
> - E.g. Figure 5 (a): around the first 100 steps. For this figure, the learning dynamics are recorded every 5 training steps.
> - Is kL < N: To achieve better OOD results, kL needs to be smaller than N. Prior work typically uses kL = N (e.g., in [1-3]).
> - Appendix B mentioned three models, but Table 3 mentions 4: we will fix this inconsistency.
> - DomainNet requires longer time to converge.
> - the ResNet18 is not really unfrozen with a large parameter k?
> The largest value k in Figure 2 or Figure 3 is always small or equals to N (**L239** in our paper) to ensure all blocks were unfrozen. To clarify, block is determined based on the implementation namespace (*L941* in our paper for ResNet). For example, the last unfreezing k in Figure 2 for MNIST, is at 800.
> - In Figure 2, you actually show the unfreezing schedule parameter K: yes, as indicated in our figure axis, captions and paragraph 1 in S5.1.
>
> Thank you for your additional suggestions on writing. We hope these clarifications address your concerns and further highlight the contributions of our study.

---

> > ### Comment · Reviewer_pcei · 2024-11-30
> >
> > Thanks, quick follow up
> >
> > >  The largest value k in Figure 2 or Figure 3 is always small or equals to N
> >
> > This does not seem to be correct according to the algorithm? kL would need to be smaller of equal to N? Could you again let me know (eg using a table) what the number of L is for every model, what the optimal k is, and how long these models were trained?
> >
> > Given your claim in the title (early phase of training) it would also need to be the case that kL is substantially smaller than N (otherwise it’s hardly the early phase).

---

> ### Author Response · Authors · 2024-11-30
> **References**
>
> 1] Kumar et al. “Fine-Tuning can Distort Pretrained Features and Underperform Out-of-Distribution.” International Conference on Learning Representations (2022).
>
> [2] Howard and Ruder. “Universal Language Model Fine-tuning for Text Classification.” Annual Meeting of the Association for Computational Linguistics (2018).
>
> [3] Raffel, Colin et al. “Exploring the Limits of Transfer Learning with a Unified Text-to-Text Transformer.” J. Mach. Learn. Res. 21 (2019): 140:1-140:67.

---

> ### Author Response · Authors · 2024-11-30
> **Re**
>
> The best kL is always substantially smaller than N, i.e., in the early period of training. All training hyperparameters (epochs or steps) are specified in Table 3. To translate the epochs into steps, here are the numbers:
>
> MNIST - optimal k around 200, training steps 4.6K
>
> CIFAR10 - optimal k around 3000, training steps 78K
>
> CIFAR100 - optimal k around 1500, training steps 78K
>
> OfficeHome - optimal k around 50, training steps 5K
>
> DomainNet -   optimal k around 200,  training steps  15K
>
> XNLI - optimal k around 1000, training steps 184K
>
> SQUAD - optimal k around 1500, training steps 49K

---

> > ### Author Response · Authors · 2024-12-04
> > **Re pcei**
> >
> > Dear Reviewer,
> > Thank you very much for the valuable discussions during the rebuttal period. If we have adequately addressed your concerns, we kindly request your consideration in revising the score.
> >
> > We truly appreciate your time and effort, thank you again.

---

### Official Review · Reviewer_4NwA · 2024-11-03

**Soundness:** 3
**Presentation:** 3
**Contribution:** 2
**Rating:** 5
**Confidence:** 3

**Summary:**

This paper investigates how gradual unfreezing the parameters during the early training stage affects OOD performances, and empirically unveils that gradual unfreezing leads to a time-sensitive trade-off between OOD and ID performance. The authors provide an explanation of the role of gradual unfreezing on OOD generalization: gradual unfreezing could help better to align early mini-batch gradients to the full-batch gradient, thus preventing potential overfitting to the mini-batches and eliminating spurious features. Moreover, they find that gradual unfreezing increases the sharpness and Fisher Information of the model parameters. The authors also propose a heuristic algorithm to find the best training step to start gradual unfreezing, gaining some empirical improvements on OOD tasks.

**Strengths:**

1. The finding of the significant impact of the early stage training on OOD performances is novel.
2. The experimental results include comprehensive OOD tasks for both vision and language domains.
3. The empirical finding that low sharpness doesn't necessarily improve OOD generalization in Figure 5 is meaningful, challenging the common belief that flat loss landscapes could benefit generalization.

**Weaknesses:**

1. Why would starting GU at the end of a dramatic change in sharpness help OOD? Since sharpness cannot reflect OOD performance, why is it used as an indicator of the timing of start GU?
2. Section 6.2 is insufficient to validate claim 2) of the hypothesis, since there lacks a clear correlation between the changes in the sharpness and the OOD performance across the training steps. It is best to plot the "sharpness-training step" curve and the "OOD acc-GU start step" curve in a single graph for all tasks and models to more clearly show the correlation between the changes in the sharpness and the best interval to start GU. Otherwise, it is hard to say that it is always best to start GU at the end of a sharp change in sharpness just by the simple MNIST experiment in line 440.
3. Table 1 and 2 merely shows the winning rate, rather than show the concrete number of improvements, which could be less convincing.

**Questions:**

Is the proposed heuristic method computationally expensive? (Because you need to train an extra time and calculate the relevant metric)

---

> ### Author Response · Authors · 2024-11-22
> **Re Reviewer 4NwA**
>
> We would like to thank reviewers for their time and feedback.
>
> W1: We appreciate the question regarding the rationale for starting gradual unfreezing (GU) at the end of a sharp change in sharpness. The decision to explore sharpness as a timing indicator is motivated by empirical observations of when GU begins to yield positive OOD results. While we do not claim that sharpness alone predicts OOD generalization, our study highlights how sharpness changes can help us identify a phase in early training where sensitivity to mini-batch gradients diminishes, thus potentially stabilizing training. This empirical finding suggests that the moment of sharp change in sharpness could serve as a practical indicator, even though sharpness itself is not a direct measure of OOD performance during training.
>
> W2: We’d like to clarify that our heuristic intervention algorithm is also for validating the hypothesis. To prevent confusion, we now revised the section of our paper related to the heuristic algorithm intervention. We updated the original “case study” section into “hypothesis validation” (L456) to be explicit.
>
> W3: We will revise these tables to include the absolute gain.
>
> Q1: Extra computation is required to track the training dynamics (the worst-case sharpness is also more costly) at least in the early period of training. We use the heuristics to arrive at the insights that change in sharpness and training dynamics as a signal to OOD generalisation, rather than prescribing it as a method. We have updated our “limitations” section (L918, appendix A) to discuss this.
>
> We hope these clarifications address your concerns and further highlight the contributions of our study.

---

> > ### Comment · Reviewer_4NwA · 2024-11-24
> >
> > Thank you for your feedback. While some of my concerns have been addressed, I still have a few additional questions:
> >
> > In my opinion, model sharpness does not necessarily determine OOD performance. However, changes in sharpness might help identify a better timing to start GU, thereby improving OOD performance. Am I understanding this correctly?
> >
> > You mentioned that "our study highlights how sharpness changes can help us identify a phase in early training where sensitivity to mini-batch gradients diminishes." Which experiment in your paper supports this claim? I may have missed it, as I couldn't find any experiment that demonstrates the correlation between changes in sharpness and sensitivity to mini-batches.
> >
> > While this paper provides novel empirical findings, I am still curious about the rationale behind them. For instance, in Section 5.2, you attribute the improvement brought by GU to avoiding overfitting to mini-batches. I find this explanation somewhat unconvincing. Even if the model avoids overfitting to specific mini-batches, it could still rely on spurious features that are prevalent across the entire training set. For example, in the Colored MNIST dataset, both training domains are dominated by the same spurious correlations between color and label.
> > Furthermore, there doesn’t seem to be a compelling explanation or analysis of why starting GU at the end of the sharpness's drastic change benefits OOD performance.
> >
> > I hope my questions will contribute to improving your work. I believe this research has the potential to be impactful by further exploring the underlying rationales behind the empirical findings.

---

> > > ### Author Response · Authors · 2024-11-24
> > > **Re 4NwA**
> > >
> > > Thank you very much for your discussion and feedback, which have been invaluable in enhancing the rigour of our paper.
> > >
> > > Re: In my opinion, model sharpness does not necessarily determine OOD performance. However, changes in sharpness might help identify a better timing to start GU, thereby improving OOD performance. Am I understanding this correctly?
> > > - Correct.
> > >
> > > Re: You mentioned that "our study highlights how sharpness changes can help us identify a phase in early training where sensitivity to mini-batch gradients diminishes." Which experiment in your paper supports this claim? I may have missed it, as I couldn't find any experiment that demonstrates the correlation between changes in sharpness and sensitivity to mini-batches.
> > > - We think the experiment in S5.2 is related to this.
> > >
> > > Re: While this paper provides novel empirical findings, I am still curious about the rationale behind them. For instance, in Section 5.2, you attribute the improvement brought by GU to avoiding overfitting to mini-batches. I find this explanation somewhat unconvincing. Even if the model avoids overfitting to specific mini-batches, it could still rely on spurious features that are prevalent across the entire training set. For example, in the Colored MNIST dataset, both training domains are dominated by the same spurious correlations between color and label. Furthermore, there doesn’t seem to be a compelling explanation or analysis of why starting GU at the end of the sharpness's drastic change benefits OOD performance.
> > > - Thank you so much for this insightful comment. Our paper was focused on reporting empirical findings. While we propose one plausible explanation for these observations, determining the true underlying reason, which likely leads to 1) development of new theory, 2) considering more spurious correlated test cases. These are definitely valuable directions for future work and natural extensions of the research.

---

> > > > ### Comment · Reviewer_4NwA · 2024-11-25
> > > >
> > > > Thank you for your further response. Could you elaborate more on how Sec. 5.2 is related to the change in sharpness? In my opinion, Sec. 5.2 only shows that GU increases the similarity between the gradients of mini-batches and that of the entire dataset. However, this alone doesn’t seem to directly reveal the impact on sharpness. Could you clarify this relationship?

---

> > > > > ### Author Response · Authors · 2024-11-30
> > > > > **Re**
> > > > >
> > > > > Thank you for the discussion. To clarify, it is correct that S5.2 only demonstrates gradient similarity with GU. Our comment is based on the observation that both high sharpness (S5.1) and high gradient similarity (S5.2) occur during the early stages of training under GU.

---

> > > > > > ### Comment · Reviewer_4NwA · 2024-12-01
> > > > > >
> > > > > > Thank you for your response. The reason why I was questioning how Sec. 5.2 is associated to the sharpness is that in your Thank you for your response. I was questioning the association between Sec. 5.2 and sharpness because, in your previous reply, you stated that the experiment in Sec. 5.2 supports your claim that "our study highlights how sharpness changes can help us identify a phase in early training where sensitivity to mini-batch gradients diminishes."
> > > > > >
> > > > > > However, I am now confused by your statement that Sec. 5.1 is responsible for this, as I couldn't find any experiments in Sec. 5.1 related to sharpness. ** For now I'm still concerned about the direct experimental evidence that support "sharpness changes can help us identify a phase in early training where sensitivity to mini-batch gradients diminishes". ** I'd appreciate if you could clarify this more detailedly.

---

> ### Author Response · Authors · 2024-12-03
> **Re**
>
> Thank you for pointing this out. We apologize for the confusion. It was a typo on our part, we intended to refer to S6.1 (instead of S5.1). In this section, we discuss the training dynamics, including sharpness.

---

> > ### Author Response · Authors · 2024-12-04
> > **Re 4NwA**
> >
> > Dear Reviewer,
> > Thank you very much for the valuable discussions during the rebuttal period. If we have adequately addressed your concerns, we kindly request your consideration in revising the score.
> >
> > We truly appreciate your time and effort, thank you again.

---

### Official Review · Reviewer_zxbt · 2024-11-04

**Soundness:** 2
**Presentation:** 3
**Contribution:** 2
**Rating:** 6
**Confidence:** 3

**Summary:**

This work investigates the impact of interventions by weight freezing during early stage of training on out of distribution generalization and reports empirical evidence on 3 different OOD tasks based on covariate shift using image and language data.
They further hypothesize that using Fisher information and sharpness measures one can detect learning dynamic phase changes to leverage this phenomenon as an effective learning algorithm.
Their analysis and results provides some evidence of the. usefulness of the proposed algorithm.

**Strengths:**

- The Discovery of the early learning dynamics and its impact on OOD generalization is an interesting finding, and the empirical evaluations seem to show that they indeed exist.
- The evaluations are done across various tasks from images and language, and via different datasets which provides generality to the observed results.
- Both FI and sharpness are well-studied topics which have been connected to generalization from both theoretical and empirical standpoints, hence, well suited for the development of the proposed method to improve OOD generalization.

**Weaknesses:**

The proposed algorithm to improve OOD generalization is --as it is also mentioned by the authors-- heuristic and does not provide deeper understanding of how these changes in learning dynamics occur or is connected to generalization, beyond what has been discovered already in the literature.The gains from the algorithm that are presented in tables 1 and 2 also seem to be marginal, and it is unclear how well they perform compared to SoTA OOD, as no real baseline has been used in this work to demonstrate how the proposed algorithm compares to latest advances in OOD generalization. Furthermore, results are provided only partially, and it is unclear how the proposed algorithm performs on domain adaptation tasks. Please provide comparisons to SoTA, and extend results to domain adaptation using the proposed algorithm. Further deepen the analysis and hypothesize why this phenomenon occurs.

**Questions:**

see the weaknesses section.

---

> ### Author Response · Authors · 2024-11-22
> **Re Reviewer zxbt**
>
> We would like to thank reviewers for their time and feedback.
>
> W1: We demonstrate empirically that when training with partial or dynamically varying parameters, sharpness in the early period of training does not predict OOD generalization. We believe this finding is interesting and important because it challenges the commonly held belief that lower sharpness throughout the entire training process is crucial for generalization. As methods involving partial or dynamic parameter training gain popularity, our findings highlight an important aspect that warrants further attention. Additional supporting cases are also detailed in our Appendix.
>
> To the best of our knowledge, the new insights provided in this work are:
>
>   - The early stages of training significantly influence OOD performance under covariate shift.
>   - The number of trainable parameters is a critical factor overlooked in previous studies.
>   - Empirical evidence shows that low sharpness or Fisher Information during the early training phase does not predict OOD generalization.
>   - Relative changes in training dynamics during the early period are more informative for OOD generalization than absolute values.
>
> W2: We would like to clarify that our study does not propose a new algorithm, we aim to provide new insights rather than to establish SOTA OOD performance. We use gradual unfreezing as a method for investigation (L20 in the abstract of our original submission). It is selected due to prior work showing its effectiveness in achieving SOTA results in various transfer learning settings [1-5]. We reflect this in our updated revision (Section 3.3).
>
> We also revised the section of our paper related to the heuristic algorithm intervention. We updated the original “case study” section into “hypothesis validation” (L456) to prevent misunderstanding. Additionally, we updated the corresponding “summary of findings” (L508) to emphasize that our intervention algorithm is for confirming insights and showing potential applications rather than rather than a prescriptive method.
>
> W3: We are happy to report preliminary results on Domain Net using the same hyperparameters specified in the paper. We included these results in the revised version of the paper (L506 and Table 7).
>
> |                              |ID / OOD / Win-Rate |
> | -------- | ------- |
> |Standard training:  |  62.77 / 35.34 / -      |
> |tr(F):                      | 63.27 / 37.86 / 90%|
> |Avg. sharpness:    |63.13 / 37.80 / 90%|
> |Worst sharpness:  |63.20 / 37.95 / 90%|
>
>
> We hope these clarifications address your concerns and emphasize our study's key contribution: demonstrating the significance of the early training phase for OOD generalization, particularly when considering trainable parameters and methods like freeze training.
>
> ---------------
> References:
>
> [1] Kumar et al. “Fine-Tuning can Distort Pretrained Features and Underperform Out-of-Distribution.” International Conference on Learning Representations (2022).
>
> [2] Lee et al. “Surgical Fine-Tuning Improves Adaptation to Distribution Shifts.” International Conference on Learning Representations (2023).
>
> [3] Liu et al.. “FUN with Fisher: Improving Generalization of Adapter-Based Cross-lingual Transfer with Scheduled Unfreezing.” North American Chapter of the Association for Computational Linguistics (2023).
>
> [4] Howard and Ruder. “Universal Language Model Fine-tuning for Text Classification.” Annual Meeting of the Association for Computational Linguistics (2018).
>
> [5] Reinhardt et al. “Improving Vision-Language Cross-Lingual Transfer with Scheduled Unfreezing.” Proceedings of the 3rd Workshop on Advances in Language and Vision Research (ALVR) (2024).

---

> > ### Comment · Reviewer_zxbt · 2024-11-25
> >
> > thank you for your response. given the additional provided results, I increase my score.
> >
> > However, the main question in the generalization literature, is how to achieve OOD generalization, and the provided empirical study does not answer that.
> >
> > some pointers in order to improve this work could include:
> > - why unfreezing increases OOD generalization? what happens exactly and analytically, and in a larger scale, that contributes to such a phenomena?
> > - how can we systematically achieve predictive OOD generalization (using unfreezing)?
> > - why (analytically, and in a larger scale) OOD increases with unfreezing? what is the main factor, and to what extend can it increase? is there a bound to this improvement? are there other contributing factor? the unfreezing by itself involves in other dimensions such as optimization, learning capacity, architectural change, learning dynamics and so on. All those aspects, and perhaps more, need to be carefully studied, as the concept of unfreezing by itself does not tell us much about OOD generalization.

---

> > > ### Author Response · Authors · 2024-11-30
> > > **Re**
> > >
> > > Thank you very much for raising the score and for your valuable suggestions. They are greatly appreciated and will be valuable for follow-up work.

---

### Author Response · Authors · 2024-11-22
**Re**

We thank the reviewers for their valuable time and feedback. We are glad to see the consensus [R1/R2/R4/R5] on the novelty and significance of our findings on the early learning dynamics and their impact on out-of-distribution (OOD) generalization. We thank the reviewer’s consensus [R1/R2/R3/R4/R5]  on the comprehensiveness of our evaluations across diverse datasets, tasks, and domains, as a convincing demonstration of the general applicability of our findings on the early learning dynamics.

We’d like to take the opportunity to reiterate on the core contributions of our paper. Our main goal was to surface novel insights on early training dynamics and their influence on OOD generalization, rather than proposing a new state-of-the-art method. We achieve this by using an existing method of gradual unfreezing. Currently, our contributions are 3-fold:

1. To the best of our knowledge, this is the first study showing how the behaviour of an early period of training impacts OOD results under covariate shift.

2. We demonstrate empirically that when training with partial or dynamically varying parameters, sharpness in the early period of training does not predict out-of-distribution (OOD) generalization. We believe this finding is interesting and important because it challenges the commonly held belief that lower sharpness throughout the entire training process is crucial for generalization. As methods involving partial or dynamic parameter training gain popularity, our findings highlight an important aspect that warrants further attention.

3. Although lower sharpness does not directly indicate generalization, sharpness patterns can signal learning *phase* changes in training. We developed the heuristic algorithm as a tool to uncover changes in training dynamics and OOD generalization, which we believe are valuable insights.

---

### Meta-Review · Area_Chair_FMad · 2024-12-16

**Metareview:**

This paper investigates the training dynamic of deep neural networks and how it impacts OOD generalization. In particular, this is the first paper that focuses on the impact of early training stages. They achieve this by gradually unfreezing, which is a method that existed before. The authors maintain that their key contribution is not the method but correlating early training dynamics with OOD generalization. The reviewers did not converge on an acceptance decision and no reviewer was willing to champion the paper. The main criticism, also acknowledged by the authors, is that the paper does not explore sufficiently how to improve OOD generalization by acting on the early phases of training. This would be important in a future resubmission to establish that the correlation the authors observed is indeed actionable to improve robustness. The reviewers mention several avenue for improvement, as:
* why unfreezing increases OOD generalization? what happens exactly and analytically, and in a larger scale, that contributes to such a phenomena?
* how can we systematically achieve predictive OOD generalization (using unfreezing)?
* why (analytically, and in a larger scale) OOD increases with unfreezing? what is the main factor, and to what extend can it increase? is there a bound to this improvement? are there other contributing factor? the unfreezing by itself involves in other dimensions such as optimization, learning capacity, architectural change, learning dynamics and so on. All those aspects, and perhaps more, need to be carefully studied, as the concept of unfreezing by itself does not tell us much about OOD generalization.

Further, reviewer pcei raised several points that the authors ignored in their reply,

**Additional Comments On Reviewer Discussion:**

The reviewers actively discussed with the authors, and while some reviewers increased their score, the concern of pcei were not addressed and I agree they should have.

---

### Decision · Program_Chairs · 2025-01-22

Reject